# The role of learning and environmental geometry in landmark-based spatial reorientation of fish (*Xenotoca eiseni*)

**Valeria Anna Sovrano**[1,2]*, **Greta Baratti**[1], **Sang Ah Lee**[3]*

1 Center for Mind/Brain Sciences (CIMeC), University of Trento, Rovereto, Italy, 2 Department of Psychology and Cognitive Science, University of Trento, Rovereto, Italy, 3 Department of Bio and Brain Engineering, Korea Advanced Institute of Science and Technology (KAIST), Daejeon, South Korea

* valeriaanna.sovrano@unitn.it (VAS); sangah.lee@kaist.ac.kr (SAL)

**Data Availability Statement:** All relevant data are within the manuscript and its Supporting Information files.

**Funding:** This work was funded by the start-up financing to VAS from CIMeC and by a research

## Abstract

Disoriented animals and humans use both the environmental geometry and visual landmarks to guide their spatial behavior. Although there is a broad consensus on the use of environmental geometry across various species of vertebrates, the nature of disoriented landmark-use has been greatly debated in the field. In particular, the discrepancy in performance under spontaneous choice conditions (sometimes called "working memory" task) and training over time ("reference memory" task) has raised questions about the task-dependent dissociability of mechanisms underlying the use of landmarks. Until now, this issue has not been directly addressed, due to the inclusion of environmental geometry in most disoriented navigation paradigms. In the present study, therefore, we placed our focus on landmark-based navigation in fish (*Xenotoca eiseni*), an animal model that has provided fruitful research in spatial reorientation. We began with a test of spontaneous navigation by geometry and landmarks (Experiment 1), showing a preference for the correct corner, even in the absence of reinforced training. We then proceeded to test landmarks without the influence of informative geometry through the use of square environments (Experiment 2–4), varying the numerosity of present landmarks, the distance of landmarks from the target corner, and the type of task (i.e., spontaneous cued memory or reference memory). We found marked differences in landmark-use in the absence of environmental geometry. In the spontaneous memory task, visual landmarks acquired perceptive salience (and attracted the fish) but without serving as a spatial cue to location when they were distal from the target. Across learning in the reference memory task, the fish overcame these effects and gradually improved in their performance, although they were still biased to learn visual landmarks near the target (i.e., as beacons). We discuss these results in relation to the existing literature on dissociable mechanisms of spatial learning.

grant to SAL from KAIST (G04170038). The funders had no role in study design, data collection and analysis, decision to publish, or preparation of the manuscript

**Competing interests:** The authors have declared that no competing interests exist.

## Introduction

Ever since the formulation of the Cognitive Map Theory [1], scientists across many disciplines have extensively questioned what the cognitive map consists of and how environmental inputs are organized within it. Important progress has been made in this endeavor in the fields of brain and cognitive sciences. One intriguing phenomenon that has been observed is the ability of disoriented rats to reorient themselves with respect to the geometric shape of the experimental environment [2, 3]. Surprisingly, rats failed in using featural cues (a wall of another color, panels with different characteristics) to disentangle the two geometrically equivalent corners, in absence of repeated experience. According to these results, Cheng [1] and Gallistel [4] proposed that a modular cognitive mechanism allows animals to spontaneously reorient by environmental geometry, followed by an experience-based learning of landmarks over time.

The general finding of geometric spatial navigation has been replicated widely in vertebrates [for reviews: 5, 6] and invertebrates [insects: 7–10]. However, there has been a debate for the past 30 years over the privileged role of environmental geometry in spatial mapping, fueled by demonstrations of successful disoriented landmark-use in conjunction with geometry in various species [for reviews: 11–14]. This variability in the use of landmarks has led to several hypotheses about competition among cues, effects of environmental size, and the role of language in humans.

In a reconceptualization of the original *modularity theory* proposed by Cheng [2] and Gallistel [4], Lee and colleagues [12, 15, 16] presented two distinct systems underlying spontaneously navigation behavior: a geometric spatial mapping system representing the extended 3D structure of the environment and an attention-dependent landmarks system representing perceptual features that directly specify particular locations. Over repeated learning, featural landmarks would also influence behavior through a route-dependent mechanism. This theoretical view, consistent with a similar view proposed by Doeller and Burgess [17], clearly distinguishes the geometric terrain layout and landmark objects as independent mechanisms for navigation, posing in contrast with alternative models to reorientation (for instance, the *view-matching* by Cheung, Stürzl, Zeil, and Cheng [18], and by Stürzl, Cheung, Cheng, and Zeil [19]; the *adaptive combination theory* by Newcombe and Huttenlocher [20]).

The view that 3D boundary structures and other kinds of features are dissociable systems is supported by evidence from a wide range of studies, from human development to function neuroimaging studies, to rodent neurophysiology studies [for a review: 21]. Recent single-cell recordings in the hippocampus of mice have revealed the importance of environmental geometry in the reorientation of hippocampal place mapping [22]. In environments containing both rectangular geometry and visual features, both the spatial reorientation behavior of mice and their hippocampal place representations were dependent on boundary geometry. Moreover, the behavior of mice was correlated, on a trial-by-trial basis, with their boundary-based hippocampal place cells' representation of location. The visual landmark cues, on the contrary, did not predict the animal's choice nor place cell mapping, even when it was presented in the absence of environmental geometry (in a square arena). Instead, features were used by mice to identify the spatial environment itself (e.g., distinguish between two rooms).

One behavioral study with mice independently tested the use of geometry and landmarks in both a working memory task and a reference memory task [23]. While geometry was used from the very beginning in both tasks, a visual landmark (i.e. striped wall) was used in a different way, depending on the task. In the working-memory task (in which the target changes on each trial) the striped wall was used only to distinguish whether the target was near striped wall without any sense of left versus right; however, with reinforcement at one stable target in

the reference-memory task, mice became increasingly accurate in identifying the target corner.

For practical reasons, spatial reorientation tasks with humans typically used something similar to a "working memory" procedure [see for example: 24]: in a single session consisting of multiple trials, the subjects had to find a corner in which they had previously observed a goal-object, without any direct reinforcement of correct choices. For children, the target is often kept in the same location in order to minimize across-trial interference (we will refer to this as a "spontaneous memory task" so as to not confuse it with traditional working-memory tests that vary the target position on each trial). Like the mice in the above study, children showed an early, consistent ability to reorient by environmental geometry but had difficulty with landmarks, only using them as beacons that mark the target location [25, 26]. Like the rats in Cheng's original study, children were often unable to integrate the environmental shape with a featural landmark [see for examples: 24, 26], in contrast with a wide range of nonhuman species, including fish, which have been shown to easily integrate geometry and landmarks in reference memory tasks, [for reviews: 5, 6, 13; in insects: see also 7–10]. Across these various studies, however, the question of task by environmental cue interaction has yet to be addressed in a single study.

## Disoriented navigation in fish

Fish have been an interesting model for spatial navigation because, despite their phylogenetic distance and difference in habitat from humans, they show functional and anatomical homologies with respect to other vertebrates. The hippocampal lateral pallium of fish has been shown to play a fundamental role in their flexible cognitive mapping ability [for a review: 27], consistent with similar findings in many other vertebrate groups–birds, reptiles, amphibians, and agnathans [for a review: 28]. For instance, in goldfish (*Carassius auratus*) it has been found that telencephalic ablation selectively corrupted allocentric spatial learning (i.e. map-like strategies) [29–33], specifically by damaging the lateral region of the telencephalic pallium [34–36]. A study by Vargas and colleagues [37] has further shown that lesions to this region severely impaired the geometry-based reorientation behavior of goldfish in a rectangular-shaped arena, leaving unaffected the use of featural cues. More recently, Rajan and colleagues [38] have demonstrated that active spatial learning in a rectangular-shaped arena induced the expression of immediate-early gene (IEG) early growth response 1 (*egr-1*) in telencephalon of goldfish, a regulatory transcription factor involved in neural plasticity and memory formation in mammals [for a review: 39]. Therefore, amniotes and teleost fishes may share the physiological mechanisms underlying the hippocampus-dependent system: indeed, Gómez and colleagues [40] have found that allocentric spatial learning of goldfish is compromised by blocking hippocampal *N-methyl-D-aspartate* (NMDA) receptors.

Early evidence of disoriented navigation in fish [41–45] showed successful learning of landmarks and geometry, but all experiments carried out with fish solely implemented a reference memory procedure: animals were trained (using reinforcement) over several days to get out through one target corner of a rectangular arena (with or without featural cues) to find food and their social conspecifics outside.

More recent studies by Lee and colleagues [46–48] implemented a novel working-memory-like task with fishes (redtail splitfin fish and zebrafish) through the use of a social motivator (i.e., viewing a conspecific in one corner) and observing subjects' preference for that corner when the conspecific is no longer present. In this task, motivated by spontaneous memory tasks with children, fish successfully used the geometry of a white rectangular arena to reorient. When presented with a conspicuous landmark (e.g., a blue wall), they chose the correct corner

when presented with the landmark in a rectangular space (thus, a geometrically informative spatial context) but with better performance when the landmark was adjacent to the target, compared to when it was far away. Although informative, it is not clear, based on these findings, which is the base for this discrepancy in landmark-use. In other words, to what extent does the environmental geometry provide a spatial context for spontaneous feature-learning and how does the proximity of the landmark to the target interact with that process? Moreover, what happens to the nature of landmark-use over the course of reinforcement learning?

## The present study

In the present study, we first investigated the use of landmarks in spontaneous choice tasks (spontaneous cued memory, non-reinforced) in both a rectangular and square environment, while varying the number and proximity of the landmarks to the target (*Experiments 1 and 2*). We then investigated the improvement in landmark-use in a reinforced reference memory task, in the absence of a geometric spatial framework (*Experiments 3 and 4*). We chose the teleost fish species *Xenotoca eiseni*, an animal model used in several reorientation studies in the past [41–44, 46].

We carried out four different experiments, comprising of several environmental or procedural conditions: Experiment 1 used a spontaneous cued memory procedure in a rectangular arena with (a) all four visual landmark panels located at the four corners; (b) two panels located at the geometrically correct corners (target corner and diagonal corner); (c) two panels located at the two geometrically incorrect corners. Experiment 2 was identical to Experiment 2 but using a geometrically uninformative square arena instead of a rectangular one. Experiment 3 considered fish using a reference memory procedure with landmarks at the corners in a square arena; Experiment 4, also used a reference memory procedure but in a square arena with a blue wall landmark, under two different conditions: a) with the target corner adjacent to the blue wall or b) with the target corner across from the blue wall.

If the use of featural landmarks in spontaneous memory is dependent on attention-based direct association rather than spatial mapping, as hypothesized in Lee and Spelke [15], only landmarks near the target should be effective. Moreover, if the environmental geometry provides the primary spatial framework during spontaneous navigation, fish may fail to reorient by landmarks (despite possible attraction to them) in the absence of a geometrically-shaped environment, replicating the results with a blue wall landmark reported in Lee et al. [46]. On the other hand, repeated experience and trained navigation in the reference memory task could strongly anchor the local cues to specific positions and/or routes, making them increasingly useful for navigation even without a geometric spatial map: in this case fish could be able to correctly solve the reorientation task in a square arena with any type of landmark (e.g., corner panels or a single blue wall), whether the reinforced corner is directly marked by the landmark or is located across from it.

## Methods

### Ethics statements

The present research was carried out in the Animal Cognition and Neuroscience Laboratory (A.C.N. Lab.) of the CIMeC (Center for Mind/Brain Sciences) at the University of Trento (Italy). All husbandry and experimental procedures complied with European Legislation for the Protection of Animals used for Scientific Purposes (Directive 2010/63/EU) and were previously authorized by the University of Trento's Ethics Committee for the Experiments on Living Organisms, and by the Italian Ministry of Health (auth. num. 1111-2015-PR).

## Subjects and housing

Subjects were 48 male mature redtail splitfin fish *Xenotoca eiseni* (ranging from 3 to 4.5 cm in body-length), from breeding stocks in our laboratory: 12 *X. eiseni* participated only in the *Experiment 1* ("Geometry + Landmarks in spontaneous memory"); 16 *X. eiseni* participated only in the *Experiment 2* ("Corner panel landmarks in spontaneous memory"); 12 X. *eiseni* participated only in the *Experiment 3* ("Landmarks in reference memory"); 8 X. *eiseni* participated only in the *Experiment 4* ("Blue wall landmark in reference memory"). Because of high motivation to rejoin female conspecifics, only males participated in experiments: in fact, females were used as social and sexual stimuli to attract the experimental subject. Fish were maintained under a 16:10-h LD cycle and kept in glass tanks (55–120 l capacity), enriched with gravel, plants and cleaned with suitable filters (Aquarium Systems Duetto 100, Newa, I) to ensure a comfortable habitat. The water temperature was maintained at 26˚C with heaters (Newa Therm) and fish were fed twice a day with dry food (GVG-Mix, Sera GmbH, D).

## Experiment 1: Geometry + Landmarks in spontaneous memory

The aim of *Experiment 1* was to understand the role of local landmarks in a rectangular space under a spontaneous cued memory procedure [46–48] in three different conditions: a) with four local landmarks, one for each corner; b) with two local landmarks, in the two corners of one diagonal of the rectangular arena, with the correct corner marked by a panel; c) with two local landmarks, in the two corners of the opposite diagonal, far from the correct corner.

**Apparatus.**   The apparatus consisted of a white rectangular tank (length: 30 cm; width: 20 cm; height: 10 cm) like the one used by Lee et al. [46].

Four panels (16 x 10 cm) were located adjacent to the corners as featural information. One panel comprised 6 white and 5 black horizontal stripes (16 cm x 0.91 cm), another one was entirely blue colored, the third panel comprised 9 green and 8 yellow vertical stripes (0.94 cm x 10 cm), the fourth one had a black X (size of stripes: 1 x 15,5 cm) (See Fig 1). Local cues were similar to that used by Sovrano et al. [42] and readapted to the new apparatus and procedure. In the condition (a) all four panels were located at the four corners; in the condition (b) only two panels were located, at the correct corner C and at the diagonally opposite corner D; in the condition (c) only two panels were located, at the two other corners (X1 and X2) on the opposite diagonal. The corner C was always the correct corner (baited with the conspecific), the corner X1 was the corner near to the correct one; the corner D was the opposite corner on the diagonal of the rectangular arena; the corner X2 was the corner far from the correct C. Different animals were observed with a different target panel (condition (a) and (b)) or with a different corner without panel (condition (c)). It should be noted that in a rectangular arena corners on each diagonal are geometrically equivalent: they have the same geometric–"metric" and "sense"–properties: for example, "a long wall on the right and a short wall on the left".

At the four corners of the arena, adjacent to the colorful panels, four small glass jars (diameter: 5 cm; depth: 6 cm), filled with water (5.9 cm in height), were placed and one of them (at the correct corner C) could host the social attractor.

In the middle of the apparatus, a transparent plastic cylinder (diameter 5 cm, height 9 cm), open at both its ends, was placed, in order to host the experimental fish, for two minutes of environmental observation before starting each trial.

The apparatus was placed in a darkened room and lit centrally from above (50 cm from the arena) by a white fluorescent light bulb (18 W; Osram GmbH, D). The apparatus rested on a turntable, which allowed the experimenter to rotate it at the end of each trial (90˚, conventionally clockwise), in order to eliminate any extra-tank cues. The height of the water was 5.9 cm, while its temperature was maintained constant at 26˚C with the aid of a heater and a filter

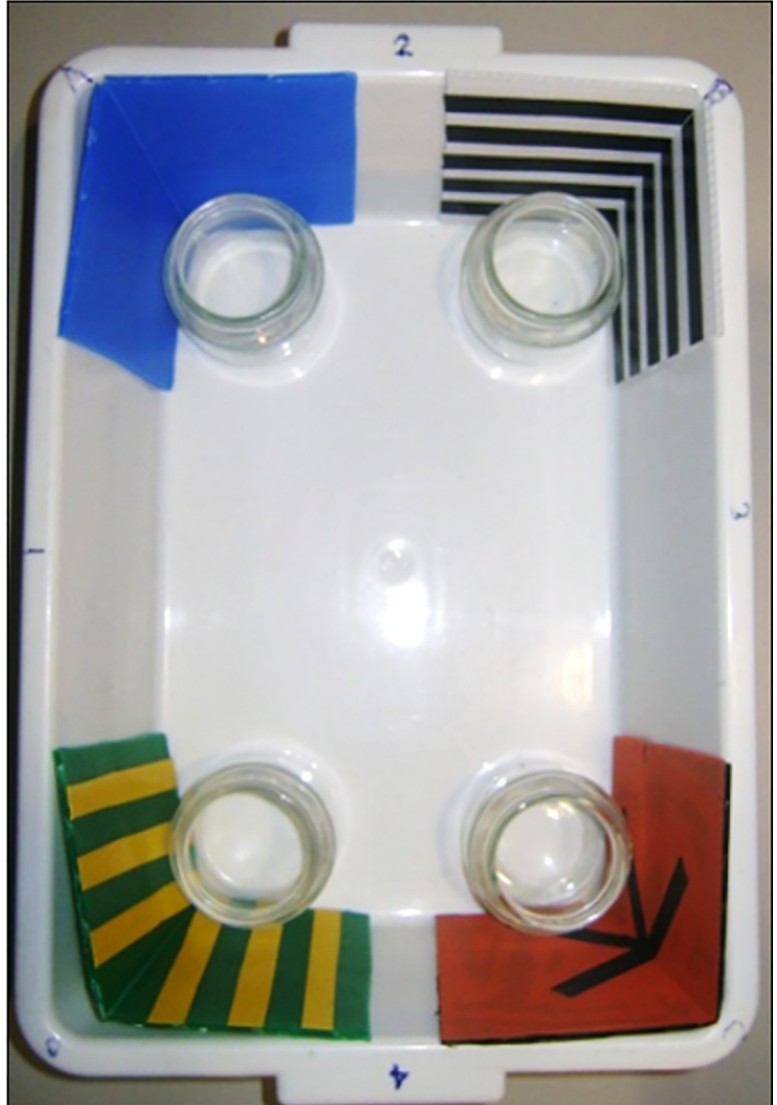

**Fig 1. Rectangular experimental arena for the spontaneous memory cued task.** Photograph of the rectangular tank used in the *Experiment 1* with different colored panels at corners, using a spontaneous memory procedure involving a social cue.

(present in not-experimental phase) ensured good water quality. A video-camera (Life Cam Studio, Microsoft, USA) recorded fish behavior.

**Procedure.** *In Experiment 1*, the procedure was similar to the one employed by Lee et al. [46–48]. At the beginning of each trial, in one of the four jars (Fig 1), the one in the reinforced corner, a female conspecific was confined, acting as a social attractor. Subsequently, the subject was placed in the center of the arena and confined in the transparent plastic cylinder. The fish had the opportunity to observe the conspecific for two minutes, without however being able to reach it (observation period). At the end of the two minutes, the cylinder containing the subject was covered with an opaque circular screen, gently carried outside the arena and rotated slowly on the turntable, 360° both clockwise and counterclockwise, in order to reduce the use of compass and inertial information (passive disorientation): the rotational movements were very gentle. Meanwhile, the jar containing the conspecific was removed and replaced with an

empty water-filled jar. The transparent plastic cylinder was then repositioned into the center of the arena and the experimental subject was gently placed inside it. The experimenter took precaution to move as carefully and as lightly as possible so as not to frighten the animals. Finally, the cylinder was lifted slowly, leaving the animal free to explore the apparatus for two minutes (test period), of which the first 30 seconds were later analyzed. At the end of the test, the animal was gently taken and the setting recomposed: the fish was placed back in the transparent plastic cylinder in the center of the arena with the conspecific social cue in one corner (the correct one C) for starting the next trial. Before each trial, the entire apparatus was rotated 90 degrees clockwise to eliminate the possible influence of any extra-tank visual cues. For the same reason, the entire apparatus was located on a table in a small darkened empty room (2 x 3 m) with homogeneous white walls.

Each trial was recorded, and the videos were subsequently coded using a grid with the "areas of choice" drawn, so that all approaches within 1 cm of the jar were considered as actual choices [46–48].

Each fish performed 12 consecutive daily trials, lasting two minutes each and with an inter-trial interval of two minutes. In each trial of test, the first approaches (the first corner approached just the fish was released) and the total approaches to the four corners within the 30 seconds after the release were considered [46–48].

An inter-observer reliability criterion [49] was applied in the re-coding of a subset of 10% of different videos ($p < 0.001$, Pearson's correlation between the ratio calculated on the original coding and on the *de novo* coding performed by an experimenter blind on the test condition of the fish).

## Experiment 2: Corner panel landmarks in spontaneous memory

Similar to *Experiment 1*, the aim of *Experiment 2* was to understand the role of local landmarks using a spontaneous memory procedure [46, 48] but in a square space, in the same three different conditions: a) with four local landmarks, one for each corner; b) with two local landmarks, in the two corners of one diagonal (C-D) of the square arena; c) with two local landmarks, in the two corners of the opposite diagonal (X1-X2), far from the correct corner C.

**Apparatus.** The apparatus was similar to that used in Lee et al. [46] and it consisted of a white square tank (25 cm x 25 cm x 10 cm) (Fig 2). The jars and the panels used were the same as in *Experiment 1*. As in the previous experiment, in the condition (a) all four panels were located at the four corners (Fig 2); in the condition (b) only two panels were located, at corners C and D; in the condition (c) only two panels were located, at the diagonally opposite corners (X1 and X2). The corner C was always the correct corner (reinforced with the social stimulus), the corner X1 was the corner on the right with respect to the correct one; the corner D was the opposite corner on the diagonal of the square arena; while the corner X2 was the corner on the left with respect to the correct C.

**Procedure.** The procedure was exactly the same as in *Experiment 1* and the animal's approaches in correspondence of the four corners were coded exactly as in *Experiment 1* (first approaches, overall number of approaches in 30 seconds of test, as well as the inter-observer reliability criterion [49] was applied).

## Experiment 3: Corner panel landmarks in reference memory

The aim of *Experiment 3* was to understand the role of local landmarks under a reference memory procedure [41, 42], again in a square space and in the same three previous conditions: a) with four local landmarks, one for each corner; b) with two local landmarks, in the two corners of one diagonal (C-D) of the square arena; c) with two local landmarks, in the two corners

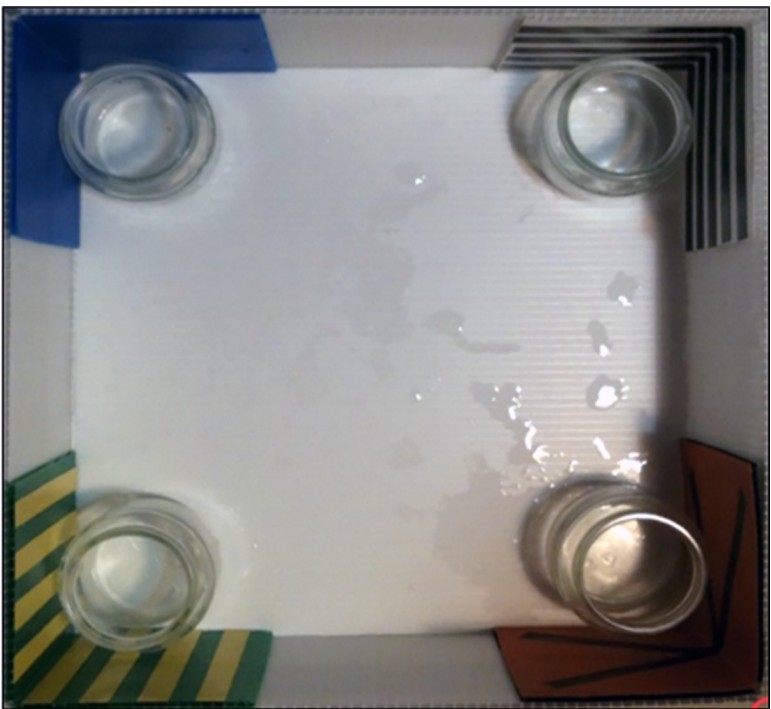

**Fig 2. Square experimental arena for the spontaneous memory task.** Photograph of square tank used in *Experiment 2* with different colored panels at corners, using a spontaneous memory procedure involving a social cue.

of the opposite diagonal (X1-X2), far from the correct corner C. For the reference memory task, fish were rewarded upon exiting the arena from the target corner with an enriched outer environment containing food and other conspecifics.

**Apparatus.** The apparatus consisted of a plastic (Poliplak®) white square arena (length: 25 cm; width: 25 cm; height: 10 cm) inserted inside a larger rectangular tank (56 cm x 38 cm x 17 cm), creating an external comfortable surrounding region, enriched with food, gravel and female conspecifics (not considered for behavioral observations), becoming an incentive to get out the experimental arena (Fig 3A). The larger rectangular tank was out of view of the

(A)

(B)

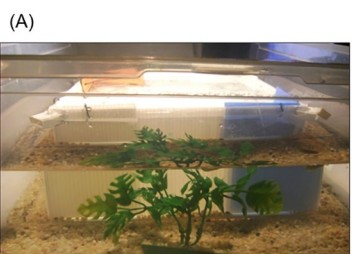
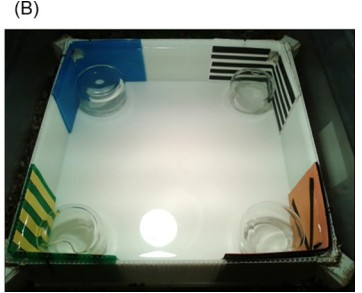

**Fig 3. Square experimental arena for the reference memory task.** (A): External view of the square apparatus used for reference memory tasks (*Experiment 3* and *4*): A plastic square tank, with four small corridors embedded in the four corners, was inserted in a larger tank and a surrounding comfortable area for fish (with gravel, plants, conspecifics and food) became an incentive to leave the apparatus. (B): Photographic enlargement of the square apparatus used for the reference memory procedure with panels (*Experiment 3*) and blue wall (*Experiment 4*): It is clearly visible the position of corridors into the corners. The presence of the glass jars kept the visual layout the same as *Experiment 2*.

experimental fish, due to the local lighting by the central fluorescent light bulb directly above the testing environment. The height of the water into the tank was 8 cm.

Embedded at the four corners of the square plastic apparatus there were four small plastic (Poliplak®) tunnels (2 cm x 3 cm, 2.5 cm in length, 6 cm from the floor; see details in Fig 3B), allowing fish to leave the transparent arena and join its conspecifics on the outside. At the end of each tunnel, there was a door (2.5 cm x 3.5 cm), made of a flexible transparent plastic material that could be easily pushed (only from the inside to the outside) and bent by the fish with its snout. Attempts to exit (choices in correspondence of corners) were considered as such only if the fish entered the tunnel along its entire body-length (2.5 cm). Only one door could be opened, other three were blocked. For the open door the flexible plastic material was glued to the perimeter of the tunnel but only sealed on the top. For blocked doors the flexible plastic material was completely sealed to the outer perimeter of the tunnel, so that it could not be opened. The upper part of each door (2.5 x 2.5 cm) was a sheet of opaque (yellow) plastic material, while the lower part (2.5 x 1 cm) was a transparent very flexible sheet. The four doors were visually identical and had also, in their lower transparent part, three equal-sized holes (0.5 cm diameter) that allowed water to pass through even the closed doors, in order to avoid potential extra-visual cues for reorientation [50]. Different animals were reinforced with a different panel at the correct door (C). The incorrect corners were coded as X1 (to the right of C), D (on the diagonal of the square arena) and X2 (on the left of C). The four jars placed at the corners (used in Experiments 1 and 2 for the spontaneous cued memory task) remained empty and completely submerged in water, in order to create the same visual set-up as in *Experiment 2*.

**Procedure.** The *Experiment 3* consisted of a training procedure of three consecutive daily sessions of 10 trials. Before starting each experimental trial, the fish was brought from the region surrounding the apparatus and gently transferred into a closed, opaque container (13 cm diameter, 7.5 cm height), and passively disoriented (slowly rotated 360° clockwise and counterclockwise) on a turntable, in order to reduce the use of compass and inertial information [41–45]. After the disorientation procedure, the fish was then gently transferred in a transparent plastic cylinder (5.5 cm diameter; 9 cm height, without top and bottom) and placed in the center of the inner tank. The animals were handled carefully and delicately, just as in *Experiments 1* and *2*.

After 10 seconds, the cylinder was removed by lifting it gently, thus leaving the fish in the center of the test-tank, free to move. In each trial, the number of choices for the four corridors was scored, until the fish was able to exit the experimental arena or at maximum for 15 minutes. If the fish made a wrong choice, it was allowed to change it until it was able to exit or until the overall time allowed for the trial elapsed [50]. Inter-trials interval, during which the fish was allowed to remain in the external region, was 10 minutes (complete reinforcement time, with also the administration of a small amount of food, presence of conspecifics and vegetation), if only the correct corner (C) was identified, and 3 minutes (reduced reinforcement time) when the correct corner was identified after two or more choices. In case of no fish response during the maximum time of the trial (15 minutes), the fish was given a pause-time of 5 minutes. Multiple choices for the correct corridor could occur, because fish explored the tunnel without actually getting out or because not enough strength was exerted to open the door. A corner attempt was considered as an effective choice if the animal entered entirely the corridor. Exit attempts were clearly visible in video-recordings through characteristic movements of the tail and the body.

The number of choices for the four corners (i.e. first choices and total number of choices per fish over the three consecutive daily sessions of 10 trials) was used as individual data. An inter-observer reliability criterion [49] was applied in the re-coding of a subset of 10% of different videos ($p<0.001$, Pearson's correlation between the ratio calculated on the original

coding and on the *de novo* coding performed by an experimenter blind on the test condition of the fish).

### Experiment 4: Blue wall landmark in reference memory

The aim of *Experiment 4* was to understand the role of a conspicuous landmark, like an entire blue wall, using the reference memory procedure [41, 42] described in Experiment 3, again in a square space and in two different conditions: a) with the correct corner C adjacent to the blue wall; b) with the correct corner C far from the blue wall.

**Apparatus.** In the *Experiment 4*, the apparatus used was exactly the same as in the previous *Experiment 3* (Fig 3) except that the corner panel landmarks were replaced with a single blue wall on one side of the arena.

**Procedure.** The procedure used for the *Experiment 4* and the individual data considered were exactly the same as in the *Experiment 3*.

### Statistical analysis

The considered variables were: In the *Experiments 1* and *2*, the mean proportions of first approaches (the first corner approached in each trial) and the overall number of approaches for each corner in 30 seconds of test; in the *Experiments 3* and *4*, the mean proportions of first choices and the overall number of choices for the four corridors per fish (until exiting the apparatus) during three consecutive daily sessions of 10 training trials.

The tests used in order to assess the homoscedasticity were the Levene's Test of equality of error variances and the Mauchly's Sphericity test. Repeated measures ANOVA was applied in order to estimate the effect of the corners and the landmarks condition, in particular the distance of the landmark from the correct corner (*Experiment 1* and *2*); the effect of the corners, the landmarks condition and the progression of learning over time (*Experiment 3* and *4*). Paired Student's t-tests were applied in order to compare frequencies of approaches (*Experiment 1* and *2*) or choices (*Experiment 3* and *4*) between pairs of corners or diagonals. To estimate the effect sizes, we reported partial eta-squared ($\eta^2_p$) as the index for ANOVA and 95% Confidence Intervals for Student's t-test. Data were analyzed with the IBM SPSS Statistics 20 software package.

### Results

#### Experiment 1: Geometry + Landmarks in spontaneous memory

In this experiment, 12 redtail splitfin fish (*Xenotoca eiseni*) were observed in a white rectangular apparatus, with four glass jars on the corners, with a social conspecific as a motivating cue at the correct corner C, and in the presence of local colorful panels. Three different experimental landmarks conditions were taken into account: (a) four panels (N = 4), (b) two panels at corners C-D (N = 4), (c) two panels at corners X1-X2 (N = 4).

The results of *Experiment 1* are shown in Figs 4 and 5.

**General analysis.** For the three experimental conditions (a), (b) and (c), data were analyzed by an ANOVA with Corners (C, D, X1, X2) as within-subjects factor and Landmarks Condition (four panels, two panels in C-D, two panels in X1-X2) as between-subjects factor.

When considering the mean proportions of first approaches at four jars on the corners (Fig 4), the analysis of variance revealed a significant effect of Corners (F(3,27) = 26.280, P ≤ 0.0001, $\eta^2_p$ = 0.745) and Corners x Landmarks Condition (F(6,27) = 10.325, P ≤ 0.0001, $\eta^2_p$ = 0.696), while the variable Landmarks Condition was not significant (F(2,9) = 1.021, P = 0.398).

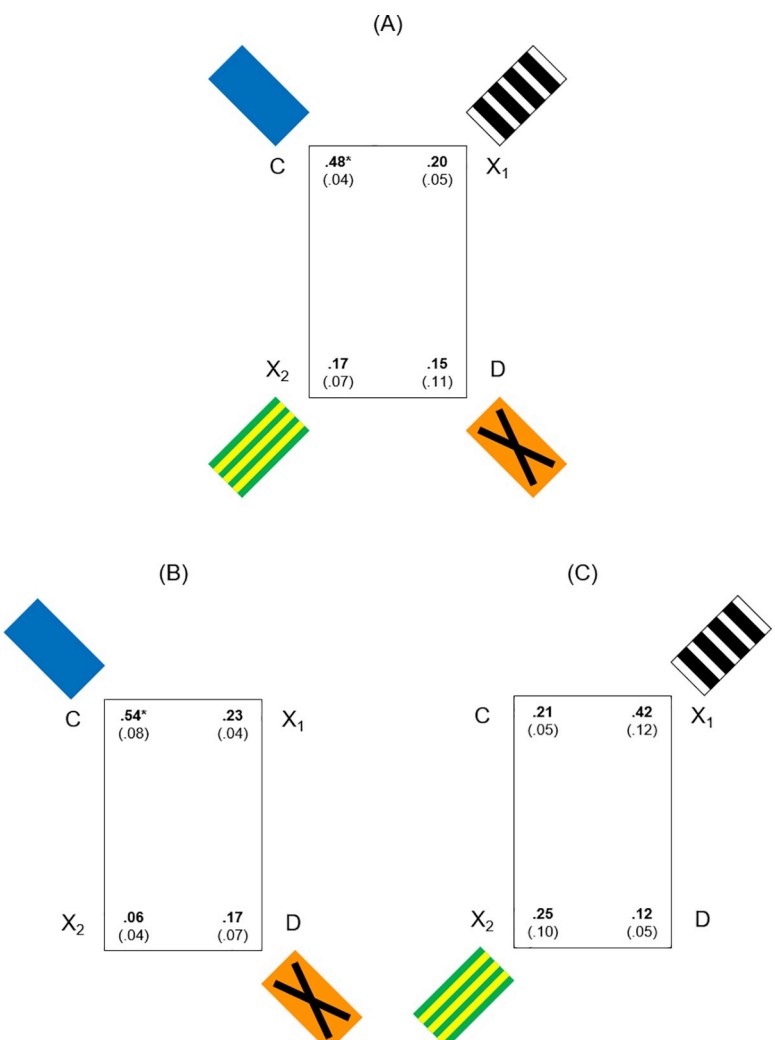

**Fig 4. First approaches in Experiment 1: Rectangular apparatus with panels in the spontaneous memory task.** (A): Mean (SD) proportions of first approaches at the four corners, in the spontaneous memory task, in the experimental condition with four colored panels at corners in the rectangular apparatus (geometry + landmarks) (*Experiment 1*). (B) and (C): Mean (SD) proportions of first approaches in the experimental conditions with two colored panels at corners C-D (landmark near the correct corner) and X1-X2 (landmark far from the correct corner) in the rectangular apparatus. The asterisk (*) indicates a statistically significant difference (p<0.05).

When considering the overall number of approaches during the spontaneous task (Fig 5), the analysis of variance revealed significant effects of Corners (F(3,27) = 21.677, P ≤ 0.0001, $\eta^2_p$ = 0.707), Corners x Landmarks Condition (F(6,27) = 4.776, P = 0.002, $\eta^2_p$ = 0.515) and Landmarks Condition (F(2,9) = 5.785, P = 0.024, $\eta^2_p$ = 0.562).

Results showed that, under a spontaneous cued memory task, in a rectangular apparatus with four panels at corners, fish did not choose randomly but they showed varying preferences in their choices, depending on the condition. In order to analyze these differences, data have been also considered separately for the three different experimental conditions.

**Detailed analysis for four panels.** In the experimental condition (a), when considering the mean proportions of first searches (Fig 4A), the analysis of variance with Corners as a within-subjects factor revealed a significant main effect of Corners (F(3,9) = 14.507, P = 0.001, $\eta^2_p$ = 0.829). A two-tailed paired t-test revealed that there were significant differences in fish

(A)

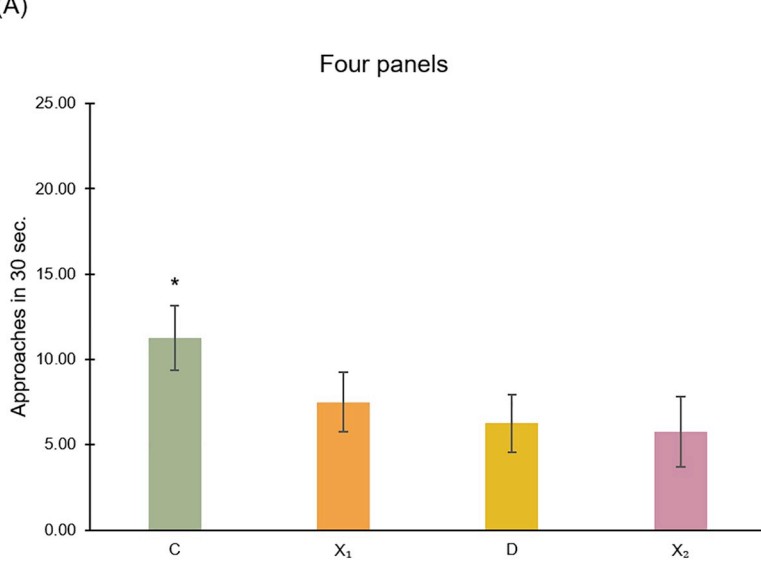

(B)

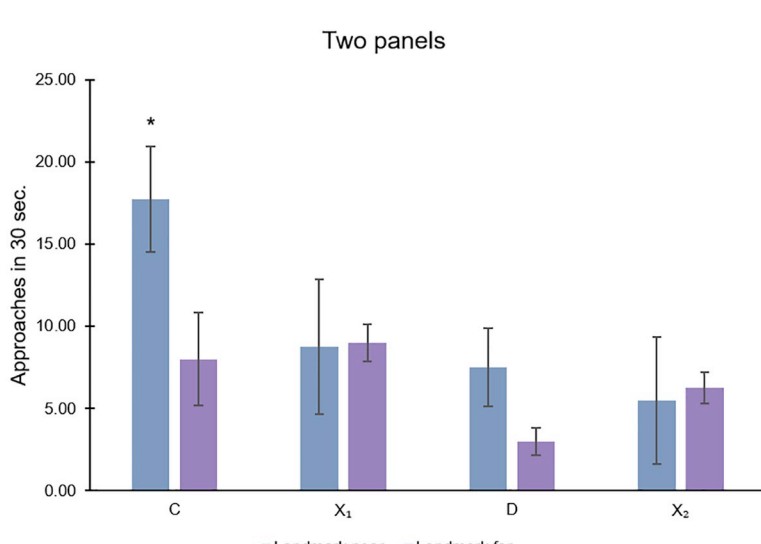

**Fig 5. Total approaches in Experiment 1: Rectangular apparatus with panels in the spontaneous memory task.**
(A): Mean (SD) number of approaches at the four corners in 30 seconds of test, under a working memory procedure, in the experimental condition with four colored panels at corners in rectangular apparatus. (B): Mean (SD) number of approaches in 30 seconds of test in the experimental conditions with two colored panels at corners C-D (landmark near the correct corner) and X1-X2 (landmark far from the correct corner). The asterisk (*) indicates a statistically significant difference (p<0.05).

choices between corners C and D ($t_{(3)}$ = 5.591, P = 0.011, 95% CI [0.143, 0.52]), C and X1 ($t_{(3)}$ = 6.789, P = 0.007, 95% CI [0.144, 0.398]), C and X2 ($t_{(3)}$ = 7.833, P = 0.004, 95% CI [0.186, 0.439]) (Fig 4A). Similar results were obtained when considering the overall number of approaches in the 30 seconds of free exploration's test (Fig 5A): the significant main effect of Corners with ANOVA (F(3,9) = 6.732, P = 0.011, $\eta^2_p$ = 0.692); significant differences in choices between the correct corner C and D ($t_{(3)}$ = 3.873, P = 0.03, 95% CI [0.891, 9.108), C

and X2 (t$_{(3)}$ = 4.261, P = 0.019, 95% CI [1.712, 9.288]), close to significance the comparison between C and X1 (t$_{(3)}$ = 2.611, P = 0.08, 95% CI [-0.82, 8.32]) (Fig 5A).

Results showed that, under a working memory procedure with the spontaneous memory task, in a rectangular apparatus with four panels at corners, fish did not choose randomly, but they directed their choices preferentially towards the correct corner C, distinguished by a peculiar panel: they seemed are able to use available landmarks for reorientation.

**Detailed analysis for two panels.** When comparing the two experimental conditions (b) and (c), the analysis of variance, applied to the mean proportions of first searches (Fig 4B and 4C), with Corners as a within-subjects factor and Distance of panels from the correct corner (panels in C-D *vs*. panels in X1-X2) as a between-subjects factor revealed a significant main effect of Corners (F(3,18) = 15.162, P ≤ 0.0001, η$^2$$_p$ = 0.716) and the interaction between Corners and Distance of panels (F(3,18) = 17.108, P ≤ 0.0001, η$^2$$_p$ = 0.74), while the Distance of panels was not significant (F(1,6) = 1.024, P = 0.351). In the condition (b), with panels in corners C-D (Fig 4B), the two-tailed paired t-test revealed significant differences in fish choices between corners C and D (t$_{(3)}$ = 5.196, P = 0.014, 95% CI [0.145, 0.605]), C and X1 (t$_{(3)}$ = 5, P = 0.015, 95% CI [0.113, 0.511]), C and X2 (t$_{(3)}$ = 12.011, P = 0.001, 95% CI [0.352, 0.606]). While in the condition (c), with panels in corners X1-X2 (Fig 4C), the two-tailed paired t-test did not reveal enough differences among corners (C-D: t$_{(3)}$ = 2.449, P = 0.092; C-X1: t$_{(3)}$ = -2.887, P = 0.063; C-X2: t$_{(3)}$ = -0.775, P = 0.495). Interesting is also the comparison between the two diagonals C-D versus X1-X2 that showed a significant difference in favor of diagonals with panels in both (b) and (c) conditions (panels in C-D, near to the correct corner: mean proportions (SD), C-D: 0.71 (0.48), X1-X2: 0.29 (0.48), t$_{(3)}$ = 8.66, P = 0.003, 95% CI [0.264, 0.57]; panels in X1-X2, far from the correct corner: mean proportions (SD), C-D: 0.33 (0.68), X1-X2: 0.67 (0.68), t$_{(3)}$ = -4.899, P = 0.016, 95% CI [-0.55, -1.117], so C-D > X1-X2 for the Near condition, while C-D < X1-X2 for the Far condition.

Similar results were obtained when considering the overall number of approaches in the 30 seconds of test (Fig 5B) with the ANOVA: a significant main effect of Corners (F(3,18) = 15.359, P ≤ 0.0001, η$^2$ = 0.719) and the interaction between Corners and Distance of panels (F(3,18) = 7.596, P = 0.002, η$^2$ = 0.559). In this case, also the Distance of panels was significant (F(1,6) = 8.435, P = 0.027, η$^2$ = 0.584). In the condition (b), with panels in corners C-D, the two-tailed paired t-test revealed significant differences in fish choices between corners C and D (t$_{(3)}$ = 4.799, P = 0.017, 95% CI [3.452, 17.048]), C and X1 (t$_{(3)}$ = 12.728, P = 0.001, 95% CI [6.749, 11.25]), C and X2 (t$_{(3)}$ = 3.927, P = 0.029, 95% CI [2.323, 22.176]). While in the condition (c), with panels in corners X1-X2, the two-tailed paired t-test revealed difference only between C and D (t$_{(3)}$ = 3.162, P = 0.05, 95% CI [-0.032, 10.032]), but not between other two corners (C-X1: t$_{(3)}$ = -1, P = 0.391; C-X2: t$_{(3)}$ = 1.578, P = 0.213).

Also with overall number of approaches of 30 seconds the comparison between the two diagonals C-D versus X1-X2 showed a significant difference in favor of diagonals with panels in both (b) and (c) conditions (panels in C-D, near to the correct corner: mean (SD), C-D: 25.25 (3.68), X1-X2: 14.25 (4.57), t$_{(3)}$ = 10.184, P = 0.002, 95% CI [7.562, 14.437]; panels in X1-X2, far from the correct corner: mean (SD), C-D: 11 (2.71), X1-X2: 17.25 (3.59), t$_{(3)}$ = -13.056, P = 0.001, 95% CI [-7.773, -4.726]). Also in this case, C-D > X1-X2 for the Near condition, while C-D < X1-X2 for the Far condition.

Results showed that, under the working memory procedure with the spontaneous cued memory task, in a rectangular arena with two panels at corners and considering the first and the total approaches, when the panel lay on the correct corner C (experimental condition (b)), fish did not choose randomly, but they directed their choices preferentially towards the correct corner C, again distinguished by a peculiar panel: they seemed are able to use available landmarks for reorientation. But when the two panels were placed far from the correct corner C, in

X1-X2 (experimental condition (c)), animals preferred the two diagonals with panels and appeared to be unable to use the landmarks to guide them to the target corner. When considering the total approaches, fish chose in equal measure the correct corner C and the two corners with panels, also in this case with a greater preference for the diagonal with landmarks. In the spontaneous memory task, fish did not seem able to use the landmarks as spatial cues to find the correct location, assembling geometry with landmarks only when they were directly associated at the target. On the other hand, the attractiveness of the panels appeared evident in both (b) and (c) conditions. These results converge with those reported in Lee and collaborators using different landmarks [46, 48].

## Experiment 2: Corner panel landmarks in spontaneous memory task

For the *Experiment* 2, 16 naïve redtail splitfin fish (*X. eiseni*) were used in the white square apparatus, with four glass jars on the corners, with a social conspecific as reward at the correct corner C, in the presence of local colorful panels. As in previous experiment, three different experimental landmarks conditions were taken into account: (a) four panels (N = 8), (b) two panels at corners C-D (N = 4), (c) two panels at corners X1-X2 (N = 4).

The results of *Experiment 2* are shown in Figs 6 and 7.

**General analysis.**   For the three experimental conditions (a), (b) and (c), data were analyzed by an ANOVA with Corners (C, D, X1, X2) as within-subjects factor and Landmarks Condition (four panels, two panels in C-D, two panels in X1-X2) as between-subjects factor.

When considering the mean proportions of first approaches at four jars on the corners (Fig 6), the analysis of variance revealed not significant effects of Corners ($F_{(3,39)}$ = 1.031, P = 0.39), Corners x Landmarks Condition ($F_{(6,39)}$ = 1.426, P = 0.229), Landmarks Condition ($F_{(2,13)}$ = 0.095, P = 0.910).

Also when considering the overall number of approaches during the spontaneous task (Fig 7), significant effects were not present (Corners: $F_{(3,39)}$ = 1.511, P = 0.227; Corners x Landmarks Condition: $F_{(6,39)}$ = 0.678, P = 0.668; Landmarks Condition: $F_{(2,13)}$ = 1.133, P = 0.352).

Results showed that, with the spontaneous cued memory task in a square apparatus and panels at corners, there were no statistically significant effects: fish chose randomly, without using available featural in any of the presented conditions.

Data have been also considered separately for the three different experimental conditions.

**Detailed analysis for four panels.**   In the experimental condition (a), when considering the mean proportions of first searches (Fig 6A), the analysis of variance with Corners as a within-subjects factor did not reveal a significant main effect of Corners ($F_{(3,21)}$ = 2.067, P = 0.135). Same results were obtained when considering the overall number of approaches in the 30 seconds of test (Corners: ($F_{(3,21)}$ = 0.535, P = 0.663) (Fig 7A).

Results showed that, under a working memory procedure with the spontaneous cued memory task, in a square apparatus with four panels at corners, fish chose randomly: they did not use the landmarks for reorientation.

**Detailed analysis for two panels.**   When comparing the two experimental conditions (b) and (c), the analysis of variance, applied to the mean proportions of first searches (Fig 6B and 6C), with Corners as a within-subjects factor and Distance of panels from the correct corner (panels in C-D *vs.* panels in X1-X2) as a between-subjects factor did not reveal significant main effects (Corners: $F_{(3,18)}$ = 0.302, P = 0.824), Corners x Distance of panels ($F_{(3,18)}$ = 2.074, P = 0.14), Distance of panels: ($F_{(1,6)}$ = 0.2, P = 0.67). Also the comparison between the two diagonals, both when panels were in C-D (condition (b)) and when they were in X1-X2

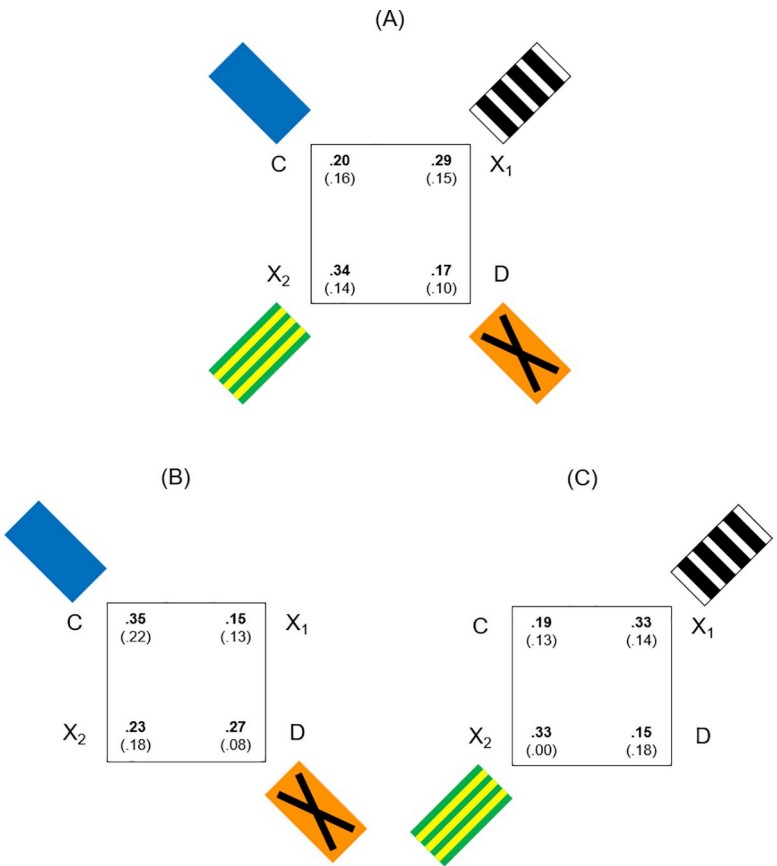

**Fig 6. First approaches in Experiment 2: Square apparatus with panels in the spontaneous memory task.** (A): Mean (SD) proportions of first approaches at the four corners, in the spontaneous cued memory task, in the experimental condition with four colored panels at corners in the square apparatus (only landmarks) (*Experiment 2*). (B) and (C): Mean proportions of first approaches in the experimental conditions with two colored panels at corners C-D (landmark near the correct corner) and X1-X2 (landmark far from the correct corner) in the square apparatus.

(condition (c)), did not show significant differences (panels in C-D: $t_{(3)} = 0.905$, P = 0.432; panels in X1-X2: $t_{(3)} = -2.449$, P = 0.092).

Similar results with the ANOVA were obtained when considering the overall number of approaches in the 30 seconds of test (Fig 7B) (Corners: $F(3,18) = 1.549$, P = 0.236), Corners x Distance of panels ($F(3,18) = 1.577$, P = 0.23), Distance of panels: ($F(1,6) = 1.47$, P = 0.271). It is worth noting the comparison between the two diagonals: in the condition (b), with panels in corners C-D (landmark near), there was no difference (mean (SD), C-D: 13.25 (6.65), X1-X2: 11.25 (4.79), $t_{(3)} = 0.463$, P = 0.675), while in the condition (c), with panels in corners X1-X2 (landmark far), the comparison between the two diagonals C-D (without panels) and X1-X2 (with panels) showed a significant difference in favor of panels far from the correct corner (mean (SD), C-D: 13 (4.69), X1-X2: 18.5 (4.2), $t_{(3)} = -4.621$, P = 0.019, 95% CI [-9.288, -1.712]) (Fig 7B).

Results showed that, in the spontaneous cued memory task in a square arena with two panels at corners (near to the goal and far from the goal), fish chose randomly and were not able to use local landmarks to guide their approaches to the correct corner. When considering the total number of approaches with panels, fish seemed to equally prefer the correct corner C and the two corners with panels even when they were far from the target. This specific result revealed the attractiveness of the panels again, as in *Experiment 1*. The strong attractiveness of

(A)

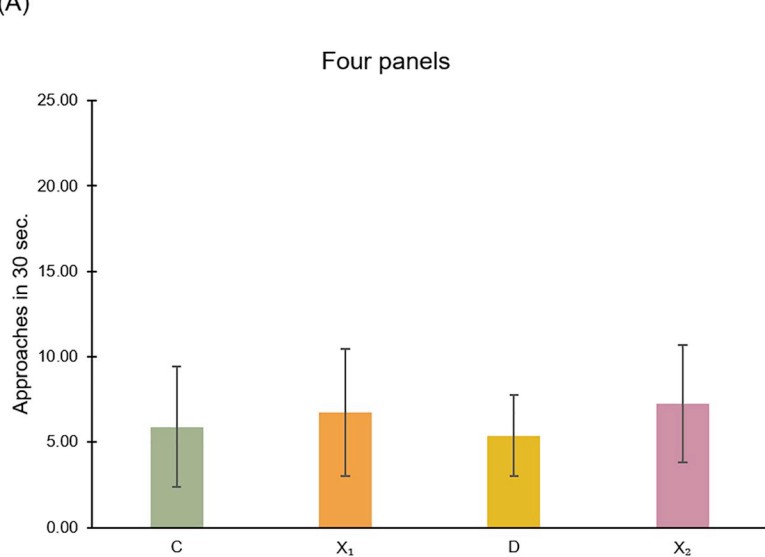

(B)

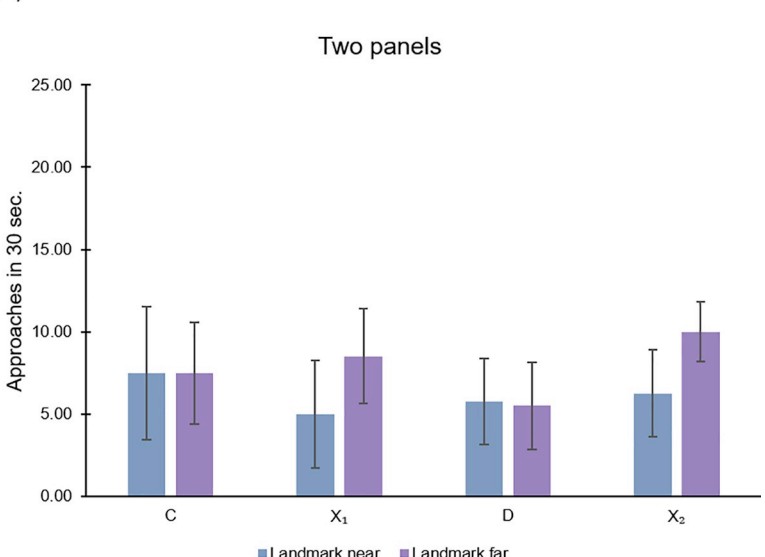

**Fig 7. Total approaches in Experiment 2: Square apparatus with panels in the spontaneous memory task.** (A): Mean (SD) number of approaches at the four corners in 30 seconds of test, in the spontaneous cued memory task, in the experimental condition with four colored panels at corners. (B): Mean (SD) number of approaches in 30 seconds of test in the experimental conditions with two colored panels at corners C-D (landmark near the correct corner) and X1-X2 (landmark far the correct corner).

the landmark in the absence of geometry had been observed also in the study of Lee and collaborators [46]. Importantly, the absence of an informative geometric structure made it more difficult to use the landmarks for spatial memory, even as associative cues.

## Experiment 3: Corner panel landmarks in the reference memory task

For the *Experiment 3*, 12 naïve redtail splitfin fish (*X. eiseni*) were used in the modified square apparatus with four corridors on the corners under a procedure of operant conditioning

(learning in locating a rewarded exit). Also in this case, the same three different experimental landmarks conditions were taken into account: (a) four panels (N = 4), (b) two panels at corners C-D (N = 4), (c) two panels at corners X1-X2 (N = 4).

The results of *Experiment 3* are shown in Fig 8.

**General analysis.** For the three experimental conditions (a), (b) and (c), data were analyzed by an ANOVA with Corners (C, D, X1, X2) and Time (three consecutive daily sessions of training) as within-subjects factors and Landmarks Condition (four panels, two panels in C-D, two panels in X1-X2) as between-subjects factor.

When considering the mean proportions of first choices at four corridors, the analysis of variance revealed a significant main effect of Corners ($F(3,27) = 52.554$, $P \leq 0.0001$, $\eta^2_p = 0.854$), Corners x Landmarks Condition ($F(6,27) = 4.692$, $P = 0.002$, $\eta^2_p = 0.51$) and Corners x Time ($F(6,54) = 5.696$, $P \leq 0.0001$, $\eta^2_p = 0.388$), while the other variables were not significant (Time: $F(2,18) = 1.003$, $p = 0.386$; Landmarks Condition: $F(2,9) = 1.021$, $P = 0.398$; Time x

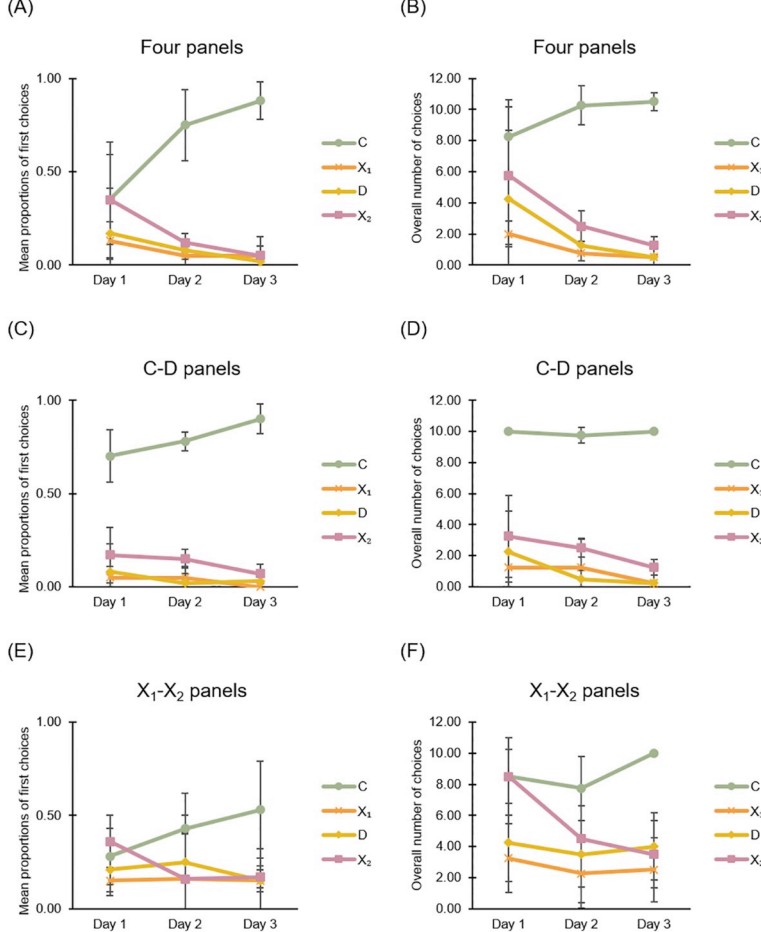

**Fig 8. Learning in Experiment 3: Reference memory task in square apparatus with panels.** (A) and (B): Graphs show mean (SD) choices at four corridors during the three daily sessions of training (reference memory for rewarded exit) in the square apparatus with four panels at corners, considering in (A) the proportions of first choices and in (B) the overall number of choices. (C) and (D): Graphs show mean (SD) choices during the three daily sessions of training in the square apparatus with two panels at corners C-D, near to the correct corner C, considering in (C) the proportions of first choices and in (D) the overall number of choices. (E) and (F): Graphs show mean (SD) choices during the three daily sessions of training in the square apparatus with two panels at corners X1-X2, far from the correct corner C, considering the proportions of first choices (E) and the overall number of choices (F).

Landmarks Condition: F(4,18) = 0.992, P = 0.437; Time x Corners x Landmarks Condition: F (12,54) = 0.795, P = 0.654).

When considering the overall number of choices during training, the analysis of variance revealed a significant main effect of Corners (F(3,27) = 82.945, P $\leq$ 0.0001, $\eta^2_p$ = 0.902), Time (F(2,18) = 5.554, P = 0.013, $\eta^2_p$ = 0.382), Corners x Time (F(6,54) = 3.615, P = 0.004, $\eta^2_p$ = 0.287) and Landmarks Condition (F(2,9) = 5.966, P = 0.022, $\eta^2_p$ = 0.57), while the other variables were not significant (Corners x Landmarks Condition: (F(6,27) = 2.317, P = 0.062; Time x Landmarks Condition: F(4,18) = 0.392, P = 0.812; Time x Corners x Landmarks Condition: F(12,54) = 0.756, P = 0.691).

Results showed that, in a reference memory task in a square apparatus with panels at corners, fish did not choose randomly but learned to direct their choices preferentially towards the correct corner C, distinguished by a specific panel, but not to the same extent in the three experimental conditions. With the aim to better understand these differences, the data were also analyzed separately for the three different experimental conditions.

**Detailed analysis for four panels.** In the experimental condition (a), in presence of four panels, data were analyzed by an ANOVA with Corners and Time (three consecutive daily sessions of learning) as within-subjects factors.

When considering the mean proportions of first choices at four corridors (Fig 8A), the analysis of variance revealed a significant main effect of Corners (F(3,9) = 12.253, P = 0.002, $\eta^2_p$ = 0.803), Corners x Time (F(6,18) = 5.603, P = 0.002, $\eta^2_p$ = 0.651), but not Time (F(2,6) = 1, P = 0.422). In the third day of learning, a two-tailed paired t-test revealed that there were significant differences in fish choices between corners C and D (t$_{(3)}$ = 13.168, P = 0.001, 95% CI [0.645, 1.097]), C and X1 (t$_{(3)}$ = 9.661, P = 0.002, 95% CI [5.532, 10.967]), C and X2 (t$_{(3)}$ = 13.113, P = 0.001, 95% CI [0.625, 1.025]). Almost similar results were obtained when considering the overall number of choices during training (Fig 8B): with the ANOVA, the significant main effect of Corners (F(3,9) = 18.289, P $\leq$ 0.0001, $\eta^2_p$ = 0.859), Time F(2,6) = 5.716, P = 0.041, $\eta^2_p$ = 0.656, but not Corners x Time (F(6,18) = 2.027, P = 0.115. Also here, but with an even greater strength than the mean proportions of first choices, in the third day of learning, a two-tailed paired t-test revealed that there were significant differences in choices between corners C and D (t$_{(3)}$ = 24.495, P $\leq$ 0.0001, 95% CI [8.701, 11.299]), C and X1 (t$_{(3)}$ = 24.495, P $\leq$ 0.0001, 95% CI [8.701, 11.299]), C and X2 (t$_{(3)}$ = 19.323, P $\leq$ 0.0001, 95% CI [7.726, 10.773])

Results showed that the choices for the correct corner C, distinguished by a specific featural panel, became particularly evident from the second training session (when considering only the mean proportions of first choices at test) and following a progressive improvement (when considering the overall number of choices) (Fig 8A and 8B).

**Detailed analysis for two panels.** In the experimental conditions (b) and (c), in presence of two panels, data were analyzed by an ANOVA with Corners and Time (three consecutive daily sessions of training) as within-subjects factors and Distance from the correct corner (panels in C-D *vs.* panels in X1-X2) as between subjects factor.

The analysis of variance, applied to the mean proportions of first choices at four corridors (Fig 8C and 8E), revealed only a significant main effect of Corners (F(3,18) = 48.002, P $\leq$ 0.0001, $\eta^2_p$ = 0.889) and Corners x Distance of panels (F(3,18) = 13.852, P $\leq$ 0.0001, $\eta^2_p$ = 0.698), while other factors and interactions were not significant (Time: F(2,12) = 1, P = 0.397; Distance: F(1,6) = 1, P = 0.356; Time x Distance: F(2,12) = 1, P = 0.397; Time x Corners: F(6,36) = 1.807, P = 0.125; Time x Corners x Distance: (F(6,36) = 0.317, P = 0.924).

During the third day of learning, in support of a successful discriminatory learning, when considering again the mean proportions of first choices at four corridors in the condition (b), with panels in corners C-D (Fig 8C), the two-tailed paired t-test revealed strong significant

differences in fish choices between corners C and D ($t_{(3)}$ = 18.278, P ≤ 0.0001, 95% CI [0.7.23, 1.027]), C and X1 ($t_{(3)}$ = 22.045, P ≤ 0.0001, 95% CI [0.77, 1.03]), C and X2 ($t_{(3)}$ = 9.66, P = 0.002, 95% CI [0.553, 1.097]). This was not equally evident in the condition (c), with panels in corners X1-X2 (Fig 8E) (two-tailed paired t-test, C-D: ($t_{(3)}$ = 2.423, P = 0.094), C-X1 ($t_{(3)}$ = 1.942, P = 0.147), C-X2 ($t_{(3)}$ = 1.732, P = 0.182) although the number of attempts in the correct corner C remained greater than in all the other corners (Fig 8E). It seems that when landmarks were far from the goal the learning appeared more difficult, becoming less prominent the corner C, and fish did not successfully learn in the three available sessions.

When considering the overall number of choices for the four corridors (Fig 8D and 8F), with the ANOVA, there were significant main effects of Corners (F(3,18) = 83.328, P ≤ 0.0001, $\eta^2_p$ = 0.933), Distance: F(1,6) = 14.833, P = 0.008, $\eta^2_p$ = 0.712; and Corners x Distance of panels (F(3,18) = 6.991, P = 0.003, $\eta^2_p$ = 0.538), while other factors and interactions were not significant (Time: F(2,12) = 2.394, P = 0.133; Time x Distance: F(2,12) = 2.906, P = 0.641; Time x Corners: F(6,36) = 2.046, P = 0.085; Time x Corners x Distance: (F(6,36) = 0.896, P = 0.508).

Also in this case, during the third day of learning, in support of a successful discriminatory learning, in the condition (b), with panels in corners C-D (Fig 8D), the two-tailed paired t-test revealed strong significant differences in fish choices between corner C and D ($t_{(3)}$ = 39, P ≤ 0.0001, 95% CI [8.954, 10.546]), C and X1 ($t_{(3)}$ = 39, P ≤ 0.0001, 95% CI [8.954, 10.546]), C and X2 ($t_{(3)}$ = 18.278, P ≤ 0.0001, 95% CI [7.226, 10.273]). This time, in the condition (c), with panels in corners X1-X2 (Fig 8F), the difference among corners was confirmed: the two-tailed paired t-test revealed difference between C and D ($t_{(3)}$ = 5.555, P = 0.012, 95% CI [2.562, 9.437]), between C and X1 ($t_{(3)}$ = 7.206, P = 0.006, 95% CI [4.188, 10.812]), C and X2 ($t_{(3)}$ = 4.181, P = 0.025, 95% CI [1.553, 11.447]). The overall number of choices, compared to only the first attempts, seemed to offer a more accurate description of the completed learning.

In a reference-memory task in a rectangular arena with two panels at corners, considering the first and the total choices, when the panel lay both on the correct corner C (experimental condition (b)) or far from the correct corner C (experimental condition (c)), fish showed to be able to learn in three consecutive days the correct position of C in relation to the positions of two remaining landmarks in the apparatus. It is also interesting that, when the two panels were disposed on the diagonal C-D, fish were already learning the discrimination from the first session of training (both considering the first and the total choices), while when the two panels were disposed on the diagonal X1-X2, far from the correct corner C, the learning began to become evident from the second daily session, with particularly pronounced results in the overall number of choices (Fig 8F). The use of panels far from the goal seemed more difficult for reorientation and this was also evident when considering the mean proportions of first choices.

## Experiment 4: Blue wall landmark in reference memory task

For this experiment, 8 naïve *X. eiseni* were used in the modified square apparatus with four corridors on the corners with a procedure of operant conditioning (training in reference memory). Two different experimental landmarks conditions were taken into account: (a) the correct corner C adjacent to the blue wall (N = 4), (b) the correct corner C distant from the blue wall (N = 4).

The results of *Experiment 4* are shown in Fig 9.

**General analysis.** For both experimental conditions (a) and (b), data were analyzed by an ANOVA with Corners and Time (three consecutive daily sessions of training) as within-

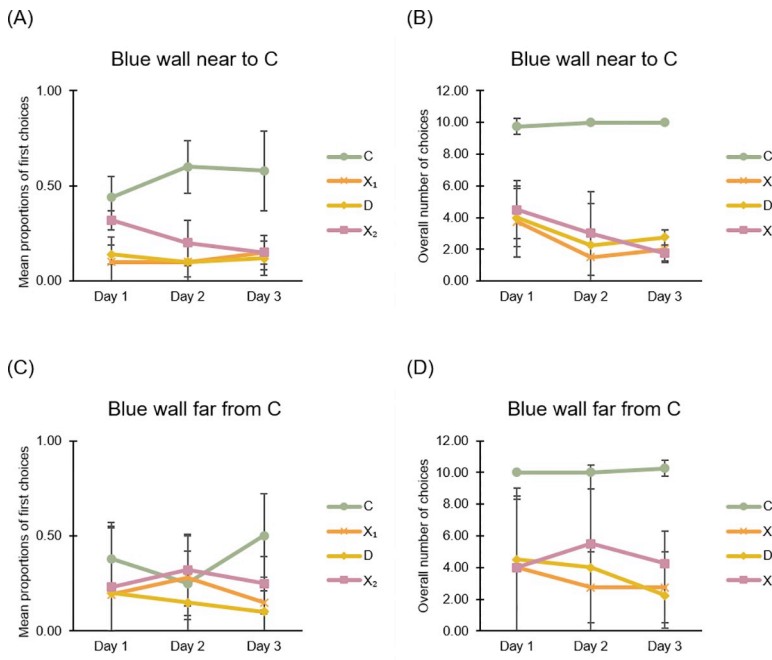

**Fig 9. Learning in Experiment 4: Reference memory task in square apparatus with blue wall.** (A) and (B): Graphs show mean (SD) choices at four corridors during the three daily sessions of training (of locating the exit through reference memory) in the square apparatus with the blue wall near to the correct corner C, considering in (A) the proportions of first choices and in (B) the overall number of choices. (C) and (D): Graphs show mean (SD) choices during the three daily sessions of training in the square apparatus with the blue wall far from the correct corner C, considering the proportions of first choices (C) and the overall number of choices (D).

subjects factors and Distance of C from the blue wall (C adjacent to the blue wall *vs.* C distant from the blue wall) as between-subjects factor.

Considering the mean proportions of first choices at the four corridors (Fig 9A and 9C), the analysis of variance revealed only a significant main effect of Corners (F(3,18) = 9.519, P = 0.001, $\eta^2_p$ = 0.613). All other factors were not significant (Time: F(2,12) $\leq$ 0.0001, P = 1; Distance: F(1,6) = 2, P = 0.207; Corners x Distance of panels (F(3,18) = 1.366, P = 0.285), Time x Distance: F(2,12) = 2, P = 0.178; Time x Corners: F(6,36) = 1.082, P = 0.391; Time x Corners x Distance: (F(6,36) = 1.658, P = 0.16).

Considering the overall number of choices at the four corridors (Fig 9B and 9D), the analysis of variance revealed only a significant main effect of Corners (F(3,18) = 36.108, P $\leq$ 0.0001, $\eta^2_p$ = 0.858). All other factors were not significant (Time: F(2,12) = 2.091, P = 0.166; Distance: F(1,6) = 0.586, P = 0.473; Corners x Distance of panels (F(3,18) = 0.246, P = 0.863), Time x Distance: F(2,12) = 0.703, P = 0.514; Time x Corners: F(6,36) = 1.243, P = 0.308; Time x Corners x Distance: (F(6,36) = 0.867, P = 0.522).

In the reference memory task, in a square apparatus with a salient landmark (an entire blue wall), fish did not choose randomly but they learned to direct their choices preferentially towards the correct corner C, distinguished by a specific relation to the blue wall.

**Detailed analysis for blue wall near to the goal.** During the third day of training, in the condition (a), with the correct corner C adjacent to the blue wall, when considering the first choices (Fig 9A), the two-tailed paired t-test revealed significant differences between corners C and D (t(3) = 3.118, P = 0.053, 95% CI [-0.009, 0.909]), C and X1 (t(3) = 3.4, P = 0.042, 95% CI [0.027, 0.822]), while the difference between C and X2 was only marginally trending toward significance (t(3) = 2.746, P = 0.071, 95% CI [-0.067, 0.917]). Nevertheless, the number of exit attempts at the correct corner C was always greater than the other three corners (Fig 9A).

Considering the total amount of choices at the four corridors (Fig 9B), the two-tailed paired t-test revealed significant differences among corner C and all other corners (C-D: $t_{(3)}$ = 29, P $\leq$ 0.0001, 95% CI [6.454, 8.046]), C-X1: $t_{(3)}$ = 19.596, P $\leq$ 0.0001, 95% CI [6.701, 9.299]), C-X2: $t_{(3)}$ = 13.113, P = 0.001, 95% CI [6.248, 10.252]).

Results showed that during the third day of training, in the condition (a), with the blue wall near the goal, the correct corner C was chosen more than the other three corners.

**Detailed analysis for blue wall far from the goal.**  During the third day of training, in the condition (b), with the correct corner C distant from the blue wall, when considering the first choices (Fig 9C), the two-tailed paired t-test revealed a significant difference between corners C and X1 ($t_{(3)}$ = 3.656, P = 0.035, 95% CI [0.045, 0.654]), but not between C and D ($t_{(3)}$ = 2.248, P = 0.11), C and X2 ($t_{(3)}$ = 1.508, P = 0.229), although the number of attempts in the correct corner C was greater than the other three corners (Fig 9C).

Considering the total number of exit attempts the four corridors, a two-tailed paired t-test revealed significant differences among corner C and all other corners (Fig 9D) (C-D: $t_{(3)}$ = 8.764, P = 0.003, 95% CI [5.095, 10.905]; C-X1: $t_{(3)}$ = 6.301, P = 0.008, 95% CI [3.712, 11.288]; C-X2: $t_{(3)}$ = 5.555, P = 0.012, 95% CI [2.562, 9.437]).

Results showed that during the third day of training, also in the condition (b), with the blue wall far from the goal, the correct corner C was chosen more than the other three incorrect corners. This effect was also evident when considering the total number of choices during 30 seconds of test.

## 4. Discussion

This work aimed to investigate the use of featural information in spontaneous memory tasks and reinforced reference memory tasks, with and without a geometric spatial framework.

Results of the *Experiment 1* showed that in the spontaneous memory task, fish directed their choices preferentially towards the correct corner and seemed able to use available landmarks for reorientation. But when the two panels were located far from the correct corner, fish did not seem able to use them as spatial cues to find the correct location, suggesting that their ability to make use of the landmark depended on a direct association of the features to the target location. In absence of an environmental geometric framework (*Experiment 2*), moreover, even the ability to use the landmarks as beacons was diminished, revealing only a strong attraction to the landmarks: fish preferred the two corners with panels but without discriminating between them to guide their approach to the target corner.

This result not only confirms previous behavioral results of Lee et al. [23] but also can be explained in terms of the reliance of hippocampal map orientations on environmental geometry rather than landmarks in disoriented mice [22]. Unlike the mice in [22], however, fish spontaneously incorporated local landmarks into their "cognitive map" when they were presented with a geometric framework (see discussion below).

When provided with reinforcement in *Experiment 3* and *Experiment 4*, fish learned to direct their choices preferentially towards the correct corner. They learned more quickly when the target was distinguished by a unique landmark (corner panels or blue wall), and when they only had to discriminate between two features rather than four. When the panel landmarks were far from the correct corner, fish had more difficulty in learning, taking three consecutive days to prefer the correct position of C in relation to the positions of the two remaining landmarks in the apparatus.

Our findings provide crucial insight regarding the unresolved debate about the inconsistencies in landmark-use in studies of reorientation. In a single study, we have shown here that while fish seem to integrate geometry and landmarks in both the spontaneous and reference

memory tasks, they are not able to fully integrate distal landmarks with the boundary geometry in absence of training. By testing only landmarks in isolation (in the square arena), we observed that the way in which the landmarks influence navigation seems to change over the course of learning, starting from direct feature association, or even simple attraction, and progressing to a relative positioning cue over time. Nevertheless, fish still show some difficulty in learning to use landmarks far from the target location.

An interesting difference in the influence of landmarks on spontaneous spatial navigation between the rectangular (*Experiment 1*) and square (*Experiment 2*) arenas is that the landmarks become attractors even though they do not allow fish to re-locate the correct corner when such features are far from it. This is particularly marked in the square arena, without the informative spatial framework provided by the boundaries. This attraction, however, disappears under the reference memory procedure. In other words, over multiple training sessions, fish learn to use features (panels: *Experiment 3*; blue wall: *Experiment 4*) as local markers to successfully reorient. Although the square and rectangular environments used in this study were equated in area, fish in the square arena may have been able to see all three corners at once and therefore, using a viewpoint-based strategy, considered all of those landmarks as part of the scene (thereby associating all of them to the target). This interpretation does not seem likely, given that there is still the problem of explaining why would be true for landmarks but not for geometry (as the same field of view would include walls of different length), as well as why a view-based scene strategy would encode each landmark separately.

It is not possible to entirely rule out the fact that there are task differences (between the spontaneous and reference memory tasks) other than the type of memory or learning. In the spontaneous choice task, the animals approach the location where a social target was last seen (but then no longer there) after an inertial disorientation, without any reinforcement of their search; in the reinforced training task, the animals learn to identify the exit one corner of the arena over multiple daily sessions, with their correct choice reinforced by a successful exit each time. Although we made every effort to make the protocol and arena comparable across the two tasks (e.g., use of same material in the visible part of the arena), there may still be motivational differences and a certain amount of novelty and surprise in the spontaneous task (with the appearance/disappearance of the conspecific). Nevertheless, the lack of learning even in the later trials of that task suggests that one of the key factors, as would be predicted, is the reinforcement provided in the reference memory task. Altogether, our results do support the existing hypothesis regarding the primacy of boundary geometry in animals' spontaneous navigation behavior–that it is processed independently from features and can be used by fish immediately, without repeated reinforcement. More importantly, our results show, for the first time in fish that environmental geometry, under spontaneous cued memory, seems to anchor the cognitive map such that landmarks can be used as direct markers to location within that map. Consequently, when the environment was square, fish could not use the landmarks to resolve the four-way spatial symmetry. This hypothesis is consistent with recent rodent place cells data regarding the role of the environmental geometry on cognitive maps [22], which show that the activity of rodents' hippocampus during disoriented navigation is driven by global environmental parameters and that the lack of a geometrically informative context causes an instability of the place cell map. According to the interpretation of the present study along these terms, when presented with a rectangular-boundary structure and some distinctive features, the place cells would have a stable map that allows for a reliable association of the features to a "place" representation on this map.

Findings from other research areas on spatial navigation corroborate the dissociation between 3D boundaries and features, too [for a review: 21]. Spatial representations of 3D boundaries and landmarks are differently implemented in the vertebrates' brain. For instance,

rodent single-neuron recording [51, 52] and both neuroimaging and direct intracranial electrophysiology in humans [17, 53] show that boundary-based navigation would be associated with the hippocampus–e.g., entorhinal cortex and subiculum–whereas landmark-based navigation would be associated with the striatum–e.g., basal ganglia. Given the strict homologies between the hippocampus of mammals and the hippocampal lateral pallium of fish [28, 54], together with a common shared pattern of the basal ganglia organization in vertebrates [55], it is reasonable that boundaries and features are dissociable spatial cues also in navigation behavior of disoriented fish [56]. Consistent with this idea, many studies on the neural correlates of behavior in fish [29–36] have shown converging evidence that hippocampus-dependent processing of allocentric spatial relationships is largely independent from cue-learning.

In conclusion, our study emphasizes the importance of the interaction between cue type (e.g., geometry vs. landmarks) and task (e.g., spontaneous spatial mapping and reinforcement-driven learning) in interpreting navigation and reorientation behavior. While the findings here are only behavioral, future studies including neural measures in the tasks and conditions that we have presented here would help test a possible dissociation at the neural level.

## Supporting information

**S1 Data.**
(XLSX)

## Acknowledgments

We wish to thank Francesco Cerri for the animal care and welfare and Vincenza Ruga for her help with experiments and data collection.

## Author Contributions

**Conceptualization:** Sang Ah Lee.

**Data curation:** Valeria Anna Sovrano.

**Formal analysis:** Valeria Anna Sovrano.

**Funding acquisition:** Sang Ah Lee.

**Methodology:** Valeria Anna Sovrano, Sang Ah Lee.

**Project administration:** Valeria Anna Sovrano, Sang Ah Lee.

**Resources:** Valeria Anna Sovrano.

**Supervision:** Valeria Anna Sovrano, Sang Ah Lee.

**Writing – original draft:** Valeria Anna Sovrano, Greta Baratti, Sang Ah Lee.

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
