## [Decision Letter · Decision Letter 0]

8 Nov 2019

PONE-D-19-26292

The role of learning and environmental geometry in landmark-based spatial navigation of fish (Xenotoca eiseni)

PLOS ONE

Ciao Valeria,

Thank you for submitting your manuscript to PLOS ONE. After careful consideration, we feel that it has merit but does not fully meet PLOS ONE’s publication criteria as it currently stands. Therefore, we invite you to submit a revised version of the manuscript that addresses the points raised during the review process.

While the reviews were generally supportive, Reviewers 1 and 3 highlighted some weaknesses in the clarity of the narrative and other weaknesses that need to be addressed in a revision. In revising your manuscript, please pay careful attention to the suggestions and criticisms of all three reviewers. I may send out your revised manuscript for re-review by Reviewers 1 and 3.

We would appreciate receiving your revised manuscript within the next 3 months. To enhance the reproducibility of your results, we recommend that if applicable you deposit your laboratory protocols in protocols.io, where a protocol can be assigned its own identifier (DOI) such that it can be cited independently in the future. For instructions see: http://journals.plos.org/plosone/s/submission-guidelines#loc-laboratory-protocols

We look forward to receiving your revised manuscript.

Kind regards,

Vern

Verner Peter Bingman, Ph.D.

Academic Editor

PLOS ONE

Journal Requirements:

1. PLOS requires an ORCID iD for the corresponding author in Editorial Manager on papers submitted after December 6th, 2016. Please ensure that you have an ORCID iD and that it is validated in Editorial Manager. To do this, go to ‘Update my Information’ (in the upper left-hand corner of the main menu), and click on the Fetch/Validate link next to the ORCID field. This will take you to the ORCID site and allow you to create a new iD or authenticate a pre-existing iD in Editorial Manager. Please see the following video for instructions on linking an ORCID iD to your Editorial Manager account: https://www.youtube.com/watch?v=_xcclfuvtxQ

Reviewers' comments:

Reviewer's Responses to Questions

**Comments to the Author**

1. Is the manuscript technically sound, and do the data support the conclusions?

Reviewer #1: Partly

Reviewer #2: Yes

Reviewer #3: Yes

2. Has the statistical analysis been performed appropriately and rigorously? 

Reviewer #1: Yes

Reviewer #2: Yes

Reviewer #3: Yes

3. Have the authors made all data underlying the findings in their manuscript fully available?

Reviewer #1: Yes

Reviewer #2: Yes

Reviewer #3: No

4. Is the manuscript presented in an intelligible fashion and written in standard English?

Reviewer #1: Yes

Reviewer #2: Yes

Reviewer #3: Yes

5. Review Comments to the Author

Reviewer #1: In the manuscript “The role of learning and environmental geometry in landmark-based spatial navigation of fish (Xenotoca eiseni)” by Sovrano et al., the authors set out to analyze the use of spatial landmarks to navigate by fish in four experiments. These four experiments evaluated spontaneous navigation by geometry and landmarks (Exp 1), spontaneous navigation by landmarks (Exp 2), the use of local landmarks under a reference memory procedure (Exp 3) and the use of a conspicuous landmark under a reference memory procedure.

In Exp 1, authors observed that, under a working memory procedure, in a rectangular apparatus with four panels at corners, fish seemed to use available landmarks for reorientation. In Exp 2, the authors reported that, under a working memory procedure, in a square apparatus with four panels at corners, fish did not use landmarks for reorientation. In Exp 3 and 4, fish trained under a reference memory procedure in a square apparatus, used either results showed that, under a reference memory procedure, in a square apparatus could use either, four panels at corners or a Wall with different color to locate the goal.

Although there are several interesting results throughout this manuscript, as a whole, it is a series of diffusely related findings that don't flow from one experiment to another. There is not really a coherent story here. It is unclear how the various studies relate to a central driving question beyond "what cues could use these fish to reorient ". Also, it is not clear whats is the contribution of this ms to the fairly amount of literatura about the use of geometry vs landmark cues.

Other comment

The methodology is sometimes confusing; some examples are:

1. They don´t explain why the use different number of subject in each experiment (12, 16, 12 and 8)

2. The absence of probe test don´t exclude the possibility that fish could be using non-spatial cues to solve the tasks. For instance, the use of feromones (the height of the water was the same that the height of the glass jars), cues related to the different exits (water flow, subtle differences in the doors/tunnels,…).

3. In Experiment 3 and 4, the fish could be using extramazes cues, for instance the geometry of the larger rectangular tank or a gradient of flow/chemical concentration.

4. The panels are 10cm wide but they comprised 11 stripes of 1cm. The same for the horizontal stripes (17 stripes of 1 cm in a panel of 16 cm)

5. Line 215 “At the four corners of the transparent arena”.

6. The “disorientation procedure” is not well explained. Do fish stay in the same water all the time? Do fish get stressed by this procedure?

7. The materials and “construction” used in the apparatus of Exp1 and Exp2 are very different and these differences could explain the differences in the result.

In the discussion they focused on the link between hippocampus and orientation based on environmental geometry and they use the available data from rodents but they did not discuss their own results with available bibliography on fish about the neural basis of the spatial navigation based on geometric cues

Reviewer #2: The article addresses reorientation behavior in fish. Specifically it examines the degree to which fish rely on landmark information in the presence of both informative and uninformative geometric cues.

The series of experiments are carefully conducted and technically sound and the paper overall is well-written and easy to follow, with the figures being especially useful. The authors employ the traditional reorientation paradigm using both rectangular and square spaces to provide informative and uninformative geometric cues to the fish. The focus here is on a type of memory being employed (working or reference memory) - given that most of the research in this field has focused on reference memory, the fact that the authors specifically address working memory makes it especially interesting since less is known about this type of behavior in the comparative literature.

The paper overall is cogent and carefully written but I did see some minor grammatical errors:

Line 58: should be "to reorient" instead of "in"

Line 112: "its" should be "their" since it's plural

Line 257: For the purpose of clarity, were there any extra-tank cues? For example, was a curtain around the apparatus or was it in an open room with furniture, etc?

Line 819: "they seemed able" (delete "are")

Line 826: "revealing only a strong attraction to the landmark" - I'm unclear which specific landmark is being referred to here or what constitutes a "strong attraction"

Line 834:"when" instead of "with"

Line 840: "dependently" - do you mean "dependent"

Line 847: Should read "they are not able to integrate"

A final point is more interpretive. The authors nicely demonstrate a difference in working memory landmark use between the rectangular space (Experiment 1) and the square space (Experiment 2) - specifically that the fish can beacon toward their correct landmark in the rectangle but not the square. As you point out in the discussion, the landmarks in the nearby corners also become attractors and the fish essentially divide their choices among 3 landmarks.

Given the reduced space of the square relative to the rectangle, is it possible that - in the absence of informative geometry - the fish are employing a viewpoint strategy in which they've encoded the 3 landmarks in close proximity of each other as a "scene" of landmarks and are treating them all as equally correct? Perhaps a point worth addressing, especially given the interest that viewpoint-matching researchers would have in this type of article.

Overall a very interesting article and a pleasure to read.

Reviewer #3: With 4 experiments, the authors examine the encoding of landmarks during reorientation in different conditions. Variables that are addressed are presence/absence of geometric information, location of landmark relative to the target, and type of paradigm (working memory or reference memory). Results confirm previous studies suggesting that the type of paradigm plays a crucial role in the encoding of landmarks.

Main issues

1) Impact. I believe the impact of the paper is not well conveyed. What makes this study different from previous studies (starting from Cheng, 1986) which have already found that feature cues are encoded less with a working memory paradigm (compared to a reference memory paradigm)? By reading this, I couldn’t understand if this study was just a replication (still worthy of reporting, but not as impactful) or something new was addressed. Related to this, I was VERY surprised that the authors didn’t cite a previous study of theirs [Lee, S. A., Tucci, V., Sovrano, V. A., & Vallortigara, G. (2015). Working memory and reference memory tests of spatial navigation in mice (Mus musculus). Journal of Comparative Psychology, 129(2), 189-197] that examines a very similar issue. From what I understand, the novelty of this study is perhaps the variable of presence/absence of useful geometry, which hasn’t received a lot of attention, rather than just WM vs RM. In any case, a more developed introduction and discussion focused on the novelty of their study is necessary to dissolve the doubts about the contribution of the present research to the literature. As it is now, this is not convincing.

2) The result section is VERY long, redundant, and difficult to follow. The authors need to improve readability. Some issues/questions/ideas are the following:

a) I would strongly suggest organizing the results more clearly in separate paragraphs, possibly with headings, and maybe with a brief conclusion FOR EACH.

b) The choice of having two DV per experiment is questionable. I am wondering if there is any advantage to reporting two DVs? They should be consistent with each other. If they are, I recommend only reporting one. If not, this is a problem – you cannot pick a DV at will in order to support an argument.

c) It also makes it confusing that you report the results BOTH comparing conditions (a, b, c) and then individually for each condition. Wouldn’t it be clearer (and shorter) just to report the analysis per each condition individually? Unless I am missing something, in that way you should still be able to infer the preference for a corner. Then you can always have a specific TARGETED comparison between conditions.

3) I don’t agree with the definition given of “working memory” procedure (Line 115) and with the way it is used in the whole paper. To the best of my knowledge, WM paradigm consist in varying the target location on each trial, such that the subject needs to use working memory to retrieve the target. This is in contrast with reference memory paradigms, in which the target location is consistent across trials and the subject can use long-term memory. Therefore, the difference between working and reference memory paradigm is NOT based on the presence of reinforcement. In fact, in Cheng’s (1986) working memory experiment, the correct location was baited (food pellets were present). In your study, did the correct corner vary on each trial? If yes, then it was a WM paradigm. It might well be that in your specific study, the task used working memory AND was also without reinforcement (free exploration); however, reinforcement and working/reference paradigm are orthogonal concepts and should not be used interchangeably. This applies to your entire paper. Please clarify the distinction and change your paper accordingly.

Minor issues

Title. The title would be more specific (and consistent with the literature) if it used the word “reorientation” rather than “navigation”.

Line 769. “while the difference between C and X2 was not COMPLETELY significant”. This wording is incorrect. Either a difference is statistically significant or not.

Experiment 3 and 4. I am wondering about the choice to use an external RECTANGULAR water tank as opposed to a square. Could this have provided geometric information to the subjects? Was the internal square plastic tank rotated relative to this larger, external, rectangular tank between trials?

Line 871. Conclusion. “fish could not use the landmarks at all to resolve the four-way spatial symmetry.” This statement needs to be qualified. They could use landmarks, at least in certain conditions.

Figures. The figures of the results should have an asterisk (*) adjacent to a significantly preferred corner. This would help a lot.

The paper needs proofreading.

Line 58. …ability of disoriented rats TO reorient…

Line 105. …has been shown to PLAY a fundamental…

Line 109. … successful learning OF landmarks and geometry…

Line 212. “remarkable to note”. Better would be “It should be noted…”

And there are MANY other cases.

6. PLOS authors have the option to publish the peer review history of their article (what does this mean?). If published, this will include your full peer review and any attached files.

Reviewer #1: No

Reviewer #2: No

Reviewer #3: No

---

## [Author Response · Author response to Decision Letter 0]

6 Feb 2020

RESPONSE TO REVIEWERS 

Reviewer #1: In the manuscript “The role of learning and environmental geometry in landmark-based spatial navigation of fish (Xenotoca eiseni)” by Sovrano et al., the authors set out to analyze the use of spatial landmarks to navigate by fish in four experiments. These four experiments evaluated spontaneous navigation by geometry and landmarks (Exp 1), spontaneous navigation by landmarks (Exp 2), the use of local landmarks under a reference memory procedure (Exp 3) and the use of a conspicuous landmark under a reference memory procedure.

In Exp 1, authors observed that, under a working memory procedure, in a rectangular apparatus with four panels at corners, fish seemed to use available landmarks for reorientation. In Exp 2, the authors reported that, under a working memory procedure, in a square apparatus with four panels at corners, fish did not use landmarks for reorientation. In Exp 3 and 4, fish trained under a reference memory procedure in a square apparatus, used either results showed that, under a reference memory procedure, in a square apparatus could use either, four panels at corners or a Wall with different color to locate the goal.

Although there are several interesting results throughout this manuscript, as a whole, it is a series of diffusely related findings that don't flow from one experiment to another. There is not really a coherent story here. It is unclear how the various studies relate to a central driving question beyond "what cues could use these fish to reorient ". Also, it is not clear whats is the contribution of this ms to the fairly amount of literatura about the use of geometry vs landmark cues.

 We thank the reviewer for this valuable feedback. We have revised and added to the introduction and discussion sections of the manuscript in order to provide a more coherent story and to emphasize the novel contribution of this work.

For instance, in the introduction, we have added the following passages:

Lines 96-103: “One behavioral study with mice independently tested the use of geometry and landmarks in both a working-memory task and a reference-memory task (Lee, Tucci, Sovrano, & Vallortigara, 2015). While geometry was used from the very beginning in both tasks, a visual landmark (i.e. striped wall) was used in a different way, depending on the task. In the working-memory task (in which the target changes on each trial) the striped wall was used only to distinguish whether the target was near striped wall without any sense of left versus right; however, with reinforcement at one stable target in the reference-memory task, mice became increasingly accurate in identifying the target corner”.

Lines 111-119: “Like the mice in the above study, children showed an early, consistent ability to reorient by environmental geometry but had difficulty with landmarks, only using them as beacons that mark the target location [25, 26]. Like the rats in Cheng’s original study, children were often unable to integrate the environmental shape with a featural landmark [see for examples: 24, 26], in contrast with a wide range of nonhuman species, including fish, which have been shown to easily integrate geometry and landmarks in reference memory tasks, [for reviews: 5, 6, 13; in insects: see also 7-10]. Across these various studies, however, the question of task by environmental cue interaction has yet to be addressed in a single study”.

The following passage is from the revised discussion section:

Lines 899-908: “Our findings provide crucial insight regarding the unresolved debate about the inconsistencies in landmark-use in studies of reorientation. In a single study, we have shown here that while fish seem to integrate geometry and landmarks in both the spontaneous and reference memory tasks, they are not able to fully integrate distal landmarks with the boundary geometry in absence of training. By testing only landmarks in isolation (in square arena), we observed that the way in which the landmarks influence navigation seems to change over the course of learning, starting from direct feature association, or even simple attraction, and progressing to a relative positioning cue over time. Nevertheless, fish still show some difficulty in learning to use landmarks far from the target location”.

Other comment

The methodology is sometimes confusing; some examples are:

1. They don´t explain why the use different number of subject in each experiment (12, 16, 12 and 8) 

The number of animals used is related to the number of experimental conditions considered in each experiment, but generally always balanced, i.e. 4 fish per experimental condition. In Experiment 1 (spontaneous cued memory, geometry + corner panel landmark), we have 3 experimental conditions (N=12); in Experiment 3 (reference memory, corner panel landmarks only) we have 3 experimental conditions (N =12); in Experiment 4 (reference memory, blue wall landmark) we have 2 experimental conditions (N=8). In Experiment 2 (spontaneous memory, corner panel landmarks only), however, there was an excess of animals available in the first experimental condition (four corner panels). Having thus collected the data, we decided to include all animals in the data analysis.

2. The absence of probe test don´t exclude the possibility that fish could be using non-spatial cues to solve the tasks. For instance, the use of feromones (the height of the water was the same that the height of the glass jars), cues related to the different exits (water flow, subtle differences in the doors/tunnels,…). 

Under the spontaneous memory procedure (the social cuing tasks), glass jars were not completely submerged in the water, so no pheromone transfer was possible: the height of the water in the glass jar was a millimeter less than its overall height, so as not to let the confined fish escape. The height of the water in the tank was also a millimeter less than the overall height of the jar. We have now clarified this in the methods sections (previous lines 216 and 225, now 247 and 256) by including the exact measurement (5.9 cm). 

Under the reference memory procedure (the exit task), possible intervening variables were checked: all corridors were visually identical, the external environment was not visible from the inside, any water flows were balanced by the presence of equal-sized holes even on the closed doors (as described at previous lines 330-333, now 368-370). The presence of more females outside the apparatus allowed a homogeneous diffusion in the environment of any pheromones. Therefore, no chemical cue would have led to the choice of one particular corner over another. Even if we assumed that it were possible somehow, it would not be able to explain the differences in behavior across environmental conditions and, in particular, the difficulty fish have in learning corners far from visual cues. Finally, a large literature on the use of this same methodology with fish (for example, Sovrano et al. 2002, Cognition; 2003, J Exp Psychol Anim B; 2005, Cognition; 2007, Anim Cogn; Sovrano & Chiandetti, 2017, Biol Communications) has clearly highlighted behavioral differences not certainly attributable to intervening variables without control, but to different types of coding of the environmental characteristics available at a given time. 

3. In Experiment 3 and 4, the fish could be using extramazes cues, for instance the geometry of the larger rectangular tank or a gradient of flow/chemical concentration.

As explained above in response to point 2, the gradient of flow and the chemical concentration were carefully considered in the design of the experiment and, therefore, kept under control.

The fish could not use extramazes cues, such as the geometry of the larger tank, because the intensity of light, centered exactly above the apparatus, in no way allowed them to see outside of the experimental tank.

For greater clarity, we have added the following sentence at previous line 315, now 351-353: “The larger rectangular tank was out of view of the experimental fish, due to the local lighting by the central fluorescent light bulb directly above the testing environment”. 

4. The panels are 10 cm wide but they comprised 11 stripes of 1cm. The same for the horizontal stripes (17 stripes of 1 cm in a panel of 16 cm) 

We thank the reviewer for noticing the error: panels comprised 17 vertical stripes, 8 yellow-9 green, 0.94 cm each one; 11 horizontal stripes, 5 black-6 white, 0. 91 cm each one. We corrected these values in the text (previous lines 198, 202, now 230, 232). 

5. Line 215 “At the four corners of the transparent arena”.

Thank you for this note. We deleted “transparent”.

6. The “disorientation procedure” is not well explained. Do fish stay in the same water all the time? Do fish get stressed by this procedure?

The “disorientation procedure” is described in detail in previous lines 244-247 (now 275-279): “the cylinder containing the subject was covered with an opaque circular screen, gently carried outside the arena and rotated slowly on a turntable, 360 degrees both clockwise and counterclockwise”. In this first phase, the fish was kept in the same container and never handled by the experimenter. The rotational movements were very gentle in order to prevent the fish from becoming distressed. After the disorientation, fish was gently placed inside a transparent plastic cylinder in the center of the apparatus. Finally the cylinder was lifted slowly, leaving the fish free to explore. The same “passive disorientation” was used under a reference memory procedure and described in lines 354-359, now 393-397. In this case the fish was brought from the region surrounding the apparatus and gently transferred in an opaque container in order to be slowly rotated, as in the previous experiments. All movements were extremely delicate so as not to frighten the animals and in order to have experimental fish in excellent condition, to be able to better measure behavioral outputs in different conditions. Our protocols, which do not cause suffering of any kind for animals, have been reviewed and approved by the University of Trento Ethics Committee for the Experiments on Living Organisms and by the Italian Ministry of Health (auth. num. 1111-2015-PR) and comply with European Legislation for the Protection of Animals used for Scientific Purposes (Directive 2010/63/EU). We have added the following sentences to meet the Reviewer's requests:

- at previous line 247, now 279: “: the rotational movements were very gentle”. 

- At previous line 250, now 282, 283: “The experimenter took precaution to move as carefully and as lightly as possible so as not to frighten the animals”.

- At previous line 359, now 399, 400: “The animals were handled carefully and delicately, just as in Experiments 1 and 2”.

7. The materials and “construction” used in the apparatus of Exp1 and Exp2 are very different and these differences could explain the differences in the result. 

The materials used in our spontanoues task were similar to those used in other papers using the same spontaneous procedures (e.g., Lee et al.,2012, Anim Cogn, Lee et al., 2015, Behav Proc). The materials used in our reference memory exit task, were similar to those used in other work by Sovrano et al. (2002, Cognition; 2003, J Exp Psychol Anim B; 2005, Cognition; 2007, Anim Cogn; Sovrano & Chiandetti, 2017, Biol Communications) using the reference memory task. In those past studies, it was shown that fish encoded a blue wall and local landmarks successfully in the presence of rectangular environmental geometry, regardless of the task and apparatus, just as we reported here in the rectangle+landmarks conditions of this paper.. In other words, just as we had different materials but the same results in the previous literature, those same differences in the apparatus now could not explain the differences in the result in the present paper. Similarly, the different materials can not explain the differences across conditions within the same task. 

Finally, the materials use for the environment itself were identical for the two tasks; the difference in the apparatus was only in the exit tunnels in the reference memory task. There is no reason to expect differences in learning performance based on those specific differences. In fact the greater complexity of the reference memory apparatus might actually lead us to predict greater difficulty in landmark learning in such an apparatus. But this was not so.

Nevertheless, we have added this passage to the discussion section, in accord with the reviewer’s concerns:

Lines 924-936: “It is not possible to entirely rule out the fact that there are task differences (between the spontaneous and reference memory tasks) other than the type of memory or learning. In the spontaneous choice task, the animals approach the location where a social target was last seen (but then no longer there) after an inertial disorientation, without any reinforcement of their search; in the reinforced training task, the animals learn to identify the exit one corner of the arena over multiple daily sessions, with their correct choice reinforced by a successful escape each time. Although we made every effort to make the protocol and arena comparable across the two tasks (e.g., use of same material in the visible part of the arena), there may still be motivational differences and a certain amount of novelty and surprise in the spontaneous task (with the appearance/disappearance of the conspecific). Nevertheless, the lack of learning even in the later trials of that task suggests that one of the key factors, as would be predicted, is the reinforcement provided in the reference memory task”. 

In the discussion they focused on the link between hippocampus and orientation based on environmental geometry and they use the available data from rodents but they did not discuss their own results with available bibliography on fish about the neural basis of the spatial navigation based on geometric cues

As requested by the Reviewer, we added and integrated into the text the existing literature on the neural basis of spatial navigation in fish:

Line 108, now lines 126-140: “For instance, in goldfish (Carassius auratus) it has been found that telencephalic ablation selectively corrupted allocentric spatial learning (i.e. map-like strategies) (Durán et al., 2008; López et al., 2000; Rodríguez et al., 2002; Salas et al., 1996a,b), specifically by damaging the lateral region of the telencephalic pallium (Broglio et al., 2010; Durán et al., 2010; Portavella & Vargas, 2005). A study by Vargas and colleagues (2006) has further shown that lesions to this region severely impaired the geometry-based reorientation behavior of goldfish in a rectangular-shaped arena, leaving unaffected the use of featural cues. More recently, Rajan and colleagues (2011) have demonstrated that active spatial learning in a rectangular-shaped arena induced the expression of immediate-early gene (IEG) early growth response 1 (egr-1) in telencephalon of goldfish, a regulatory transcription factor involved in neural plasticity and memory formation in mammals (for a review: Knapska & Kaczmarek, 2004). Therefore, amniotes and teleost fishes may share the physiological mechanisms underlying the hippocampus-dependent system: indeed, Gómez and colleagues (2006) have found that allocentric spatial learning of goldfish is compromised by blocking hippocampal N-methyl-D-aspartate (NMDA) receptors.

Line 895, now lines 963-965: Consistent with this idea, many studies on the neural correlates of behavior in fish (Broglio et al., 2010; Durán et al., 2008; 2010; López et al., 2000; Portavella & Vargas, 2005; Rodríguez et al., 2002; Salas et al., 1996a,b) have shown converging evidence that hippocampus-dependent processing of allocentric spatial relationships is largely independent from cue-learning 

Reviewer #2: The article addresses reorientation behavior in fish. Specifically it examines the degree to which fish rely on landmark information in the presence of both informative and uninformative geometric cues.

The series of experiments are carefully conducted and technically sound and the paper overall is well-written and easy to follow, with the figures being especially useful. The authors employ the traditional reorientation paradigm using both rectangular and square spaces to provide informative and uninformative geometric cues to the fish. The focus here is on a type of memory being employed (working or reference memory) - given that most of the research in this field has focused on reference memory, the fact that the authors specifically address working memory makes it especially interesting since less is known about this type of behavior in the comparative literature.

The paper overall is cogent and carefully written but I did see some minor grammatical errors:

Line 58, now 56: should be "to reorient" instead of "in"

Line 112, now 145: "its" should be "their" since it's plural

Thank you. We have made the above corrections.

At previous lines 111,112 (now 143-145) we also reformulated the sentence in the following way: “animals were trained (using reinforcement) over several days to get out through one target corner of a rectangular arena (with or without featural cues) to find food and their social conspecifics outside”. Moreover, in order to maintain textual clarity and continuity, we deleted the previous paragraph at lines 112-123: “through a target corner from a rectangular arena using its geometrical characteristics or integrating the environmental geometry with landmarks. Spatial reorientation tasks with human adults and children, on the other hand, used a “working memory” procedure with spontaneous choices [see for example: 30]: the subject had to find a corner in which he had previously observed the presence of a goal-object in a single daily session of a number of trials, without directly reinforcing correct choices. There was no continued learning over time. Like the rats in similar working memory tasks, children were not able to integrate the environmental shape with a featural landmark [see for examples: 30, 31]. Under reference memory conditions, instead, different species of animals were able to easily integrate the two different spatial information (geometric and featural) [for reviews: 5, 13; in insects: see also 7, 10]”.

Line 257: For the purpose of clarity, were there any extra-tank cues? For example, was a curtain around the apparatus or was it in an open room with furniture, etc?

In order to minimize the influence of each extra-tank visual cues the apparatus was located in a small empty room (2 x 3 m) with homogeneous white walls. A table supported the apparatus. A camera recorded from above the lamp which illuminated the tank, making it not visible (due to the light). Nevertheless, in order to ensure that there were no other uncontrolled cues, we also implemented a rotation procedure following disorientation on each trial. In any case, for greater clarity, in previous line 257 (now 290, 291), we added the following sentence: “For the same reason, the entire apparatus was located on a table in a small darkened empty room (2 x 3 m) with homogeneous white walls”.

Line 819, now 878: "they seemed able" (delete "are")

We made this change. Thank you very much.

Line 826: "revealing only a strong attraction to the landmark" - I'm unclear which specific landmark is being referred to here or what constitutes a "strong attraction"

We were referring to the preference for the two diagonals with panels/landmarks in general (not a specific panel/landmark), without the ability to distinguish between those panels/landmarks. For greater clarity at this point in the text, we have modified the sentence as follows (lines 884-886): “revealing only a strong attraction to the landmarks: fish preferred the two corners with panels but without discrimating between them to guide their approach to the target corner”.

Line 834, now 895:"when" instead of "with"

Line 840: "dependently" - do you mean "dependent"

Line 847: Should read "they are not able to integrate"

Thank you for these corrections. For previous lines 840, 847, we reformulated the paragraph (lines 899-908).

A final point is more interpretive. The authors nicely demonstrate a difference in working memory landmark use between the rectangular space (Experiment 1) and the square space (Experiment 2) - specifically that the fish can beacon toward their correct landmark in the rectangle but not the square. As you point out in the discussion, the landmarks in the nearby corners also become attractors and the fish essentially divide their choices among 3 landmarks.

Given the reduced space of the square relative to the rectangle, is it possible that - in the absence of informative geometry - the fish are employing a viewpoint strategy in which they've encoded the 3 landmarks in close proximity of each other as a "scene" of landmarks and are treating them all as equally correct? Perhaps a point worth addressing, especially given the interest that viewpoint-matching researchers would have in this type of article.

Overall a very interesting article and a pleasure to read.

We thank the reviewer for this interesting point. Although we did equate for the total size of the square and rectangular arenas (rectangular apparatus 30cm x 25cm x 10cm); square apparatus 25cm x 25cm x 10cm), it is still important to bring up the issue of visual image-matching in relation to our findings. We have added the following passage to the discussion section (Lines 916-923): 

“Although the square and rectangular environments use in this study were equated in area, fish in the square arena may have been able to see all three corners at once and therefore, using a viewpoint-based strategy, considered all of those landmarks as part of the scene (thereby associating all of them to the target). This interpretation does not seem likely, given that there is still the problem of explaining why would be true for landmarks but not geometry (as the same field of view would include walls of different length), as well as why a view-based scene strategy would encode each landmark separately”.

Reviewer #3: With 4 experiments, the authors examine the encoding of landmarks during reorientation in different conditions. Variables that are addressed are presence/absence of geometric information, location of landmark relative to the target, and type of paradigm (working memory or reference memory). Results confirm previous studies suggesting that the type of paradigm plays a crucial role in the encoding of landmarks.

Main issues

1) Impact. I believe the impact of the paper is not well conveyed. What makes this study different from previous studies (starting from Cheng, 1986) which have already found that feature cues are encoded less with a working memory paradigm (compared to a reference memory paradigm)? By reading this, I couldn’t understand if this study was just a replication (still worthy of reporting, but not as impactful) or something new was addressed. Related to this, I was VERY surprised that the authors didn’t cite a previous study of theirs [Lee, S. A., Tucci, V., Sovrano, V. A., & Vallortigara, G. (2015). Working memory and reference memory tests of spatial navigation in mice (Mus musculus). Journal of Comparative Psychology, 129(2), 189-197] that examines a very similar issue. From what I understand, the novelty of this study is perhaps the variable of presence/absence of useful geometry, which hasn’t received a lot of attention, rather than just WM vs RM. In any case, a more developed introduction and discussion focused on the novelty of their study is necessary to dissolve the doubts about the contribution of the present research to the literature. As it is now, this is not convincing.

We would like to thank the reviewer for this extremely helpful comment. The reviewer is absolutely right about the framing of these findings in relation to the past literature and the need to emphasize their impact. We have revised the manuscript accordingly, throughout. In particular, we have revised the introduction and discussion. (Please refer to our response to the first point of Reviewer 1 for the revised passages.)

2) The result section is VERY long, redundant, and difficult to follow. The authors need to improve readability. Some issues/questions/ideas are the following:

a) I would strongly suggest organizing the results more clearly in separate paragraphs, possibly with headings, and maybe with a brief conclusion FOR EACH.

We changed the results accordingly to the Reviewer’s suggestion, organizing them in separate paragraphs and brief conclusions with the following headings: for the Exp. 1, Exp. 2 and Exp. 3: “General analysis”, “Detailed analysis for four panels”, “Detailed analysis for two panels”; for the Exp. 4: “General analysis”, “Detailed analysis for blue wall near to the goal”, Detailed analysis for blue wall far from the goal”.

b) The choice of having two DV per experiment is questionable. I am wondering if there is any advantage to reporting two DVs? They should be consistent with each other. If they are, I recommend only reporting one. If not, this is a problem – you cannot pick a DV at will in order to support an argument.

The choice to consider one DV over another could be strongly linked to the type of behavioral task used. It may make sense to consider the first choice in a single session of the social cuing task, which involves a cued spontaneous memory at each test, while it may make sense to consider all the choices made up to the solution of a task, such as exiting an experimental apparatus, evaluated on several consecutive days, where there is or begins to be a consolidation in memory.

In the past, we have considered only the overall choices, but many reviewers asked to indicate the first choices as well. For this reason, we have decided to keep both DVs, to meet any potential request from an external reader. This choice also allows us to compare the two DVs within the same behavioral task (spontaneous memory task or reference memory task) and between different behavioral tasks. 

c) It also makes it confusing that you report the results BOTH comparing conditions (a, b, c) and then individually for each condition. Wouldn’t it be clearer (and shorter) just to report the analysis per each condition individually? Unless I am missing something, in that way you should still be able to infer the preference for a corner. Then you can always have a specific TARGETED comparison between conditions.

We understand the reviewer’s point that it would be simpler to report the analysis per each condition individually. However, given that a standard statistical procedure requires that we start with an overall statistical analysis and when there are significances proceed with more detailed analysis of the separate conditions, we have chosen to keep those overall comparisons as well, in order to be conservative with our results. It is true that this contributes to the length and complexity of the results section; for this reason we have chosen to reorganize it in sub-paragraphs, as recommended by the Reviewer in the previous point 2a) and hope that this reorganization improves the readability of this part of the paper. It is also true that we have conducted and reported many comparisons, both significant and non-significant: we have chosen to do this because, without them, the readers may be left to wonder whether two corners (even if they are not the target corners) are chosen with equal/different degree of preference. 

3) I don’t agree with the definition given of “working memory” procedure (Line 115) and with the way it is used in the whole paper. To the best of my knowledge, WM paradigm consist in varying the target location on each trial, such that the subject needs to use working memory to retrieve the target. This is in contrast with reference memory paradigms, in which the target location is consistent across trials and the subject can use long-term memory. Therefore, the difference between working and reference memory paradigm is NOT based on the presence of reinforcement. In fact, in Cheng’s (1986) working memory experiment, the correct location was baited (food pellets were present). In your study, did the correct corner vary on each trial? If yes, then it was a WM paradigm. It might well be that in your specific study, the task used working memory AND was also without reinforcement (free exploration); however, reinforcement and working/reference paradigm are orthogonal concepts and should not be used interchangeably. This applies to your entire paper. Please clarify the distinction and change your paper accordingly.

The two behavioral tasks are procedurally similar but different in terms of the required memory processes: on the one hand, a social cuing task in a single session, which involves a spontaneous, un-reinforced memory at each trial, in the presence of strong motivation to find a conspecific that disappeared during the test, and, on the other hand, a task that requires a solution to exiting an experimental apparatus (without a cued reminder like in the first task), offering an opportunity (multiple reinforcement contingencies) on each trial and over several consecutive days.

It is true that our spontaneous memory task (social cued task) is similar to the “working memory” tasks in the past in that we do not reward the correct choices to shape learning across trials. However, as the reviewer points out, we have chosen to use the same target corner across trials in order to minimize interference across trials, a practice often seen in developmental studies with young children. Therefore, we will refer to our task as a spontaneous memory task (rather than a working memory task) to emphasize the lack of reward for specific behavioral searches. 

We now also explain and clarify our terminology starting from the introduction:

Lines 104-110: “For practical reasons, spatial reorientation tasks with humans typically use something similar to a “working memory” procedure [see for example: 30]: in a single session consisting of multiple trials, the subjects have to find a corner in which they had previously observed a goal-object, without any direct reinforcement of correct choices. For children, the target is often kept in the same location in order to minimize across-trial interference (we will refer to this as a “spontaneous memory task” so as to not confuse it with traditional working-memory tests that vary the target on each trial).

Minor issues

Title. The title would be more specific (and consistent with the literature) if it used the word “reorientation” rather than “navigation”.

Thank you. We have made this change.

Line 769, now 833. “while the difference between C and X2 was not COMPLETELY significant”. This wording is incorrect. Either a difference is statistically significant or not.

We changed “not completely” to “only marginally trending toward to significance”.

Experiment 3 and 4. I am wondering about the choice to use an external RECTANGULAR water tank as opposed to a square. Could this have provided geometric information to the subjects? Was the internal square plastic tank rotated relative to this larger, external, rectangular tank between trials?

The rectangular tank outside was completely out of view from the fish. The intense central light locally illuminated only the square apparatus in the centre of the rectangular tank and the intensity of its light would have made extremely difficult to see or to recognize extra-tank cues.

In any case, for clarity, we added the following sentence at previous line 315, now lines 351-353: “The larger rectangular tank was out of view of the experimental fish, due to the local lighting by the central fluorescent light bulb directly above the testing environment”. 

Line 871. Conclusion. “fish could not use the landmarks at all to resolve the four-way spatial symmetry.” This statement needs to be qualified. They could use landmarks, at least in certain conditions. 

We qualified the statement, as suggested by the Reviewer, deleting “at all” and adding the specification at previous line 868, now 939-941: “More importantly, our results show, for the first time in fish that environmental geometry, under spontaneous cued memory, seems to anchor the cognitive map …”. 

Figures. The figures of the results should have an asterisk (*) adjacent to a significantly preferred corner. This would help a lot.

We made this change; thank you for the suggestion.

The paper needs proofreading.

Line 58, now line 56. …ability of disoriented rats TO reorient…

Line 105, now 124. …has been shown to PLAY a fundamental…

Line 109, now 141. … successful learning OF landmarks and geometry…

Line 212, now 242. “remarkable to note”. Better would be “It should be noted…”

And there are MANY other cases.

Thank you for pointing out these errors. We have made these corrections as suggested.

Moreover, we added several other corrections (tracked in red in the revised manuscript with track changes), in order to improve the text quality.

---

## [Editor Report · Decision Letter 1]

11 Feb 2020

The role of learning and environmental geometry in landmark-based spatial reorientation of fish (Xenotoca eiseni)

PONE-D-19-26292R1

Dear Dr. Sovrano,

We are pleased to inform you that your manuscript has been judged scientifically suitable for publication and will be formally accepted for publication once it complies with all outstanding technical requirements.

With kind regards,

Verner Peter Bingman, Ph.D.

Academic Editor

PLOS ONE
---

## [Editor Report · Acceptance letter]

18 Feb 2020

PONE-D-19-26292R1 

The role of learning and environmental geometry in landmark-based spatial reorientation of fish (*Xenotoca eiseni*)

Dear Dr. Sovrano:

I am pleased to inform you that your manuscript has been deemed suitable for publication in PLOS ONE. Congratulations! Your manuscript is now with our production department. 

With kind regards,

on behalf of

Dr. Verner Peter Bingman 

Academic Editor

PLOS ONE